# Vulnerability and tuberculosis treatment outcomes in urban settings in England: A mixed-methods study

**Luis C. Berrocal-Almanza**[1], **Marcela Lima**[1], **Helen Piotrowski**[1], **Julie Botticello**[2], **Amarjit Badhan**[1], **Nisha Karnani**[1], **Hanna Kaur**[3], **Manish Pareek**[4], **Pranabashis Haldar**[5], **Martin Dedicoat**[3], **Onn Min Kon**[1,6], **Dominik Zenner**[1,7], **Ajit Lalvani**[1,8]*

1 National Heart and Lung Institute, NIHR Health Protection Research Unit in Respiratory Infections, Imperial College London, London, United Kingdom, 2 Department of Allied and Public Health, School of Health, Sport and Bioscience, University of East London, London, United Kingdom, 3 Birmingham & Solihull TB Service, University Hospitals Birmingham NHS Foundation Trust, Birmingham, United Kingdom, 4 Department of Respiratory Sciences, University of Leicester, Leicester, United Kingdom, 5 Department of Respiratory Sciences, Institute for Lung Health, Respiratory Biomedical Research Centre, University of Leicester, Leicester, United Kingdom, 6 St Mary's Hospital, Imperial Healthcare NHS Trust, London, United Kingdom, 7 Wolfson Institute for Population Health, Queen Mary University, London, United Kingdom, 8 National Heart and Lung Institute, Tuberculosis Research Centre, Respiratory Medicine, Imperial College London, London, United Kingdom

* a.lalvani@imperial.ac.uk

**Data Availability Statement:** All relevant data are within the paper and its Supporting Information files.

## Abstract

### Background

Evidence on factors contributing to poor treatment outcome and healthcare priorities in vulnerable populations affected by tuberculosis (TB) in urban areas of England other than London is needed to inform setting-specific prevention and care policies. We addressed this knowledge gap in a cohort of TB patients and healthcare providers in Birmingham and Leicester, UK.

### Methods

A mixed-methods study was performed. Logistic regression was used to identify TB patients more likely to have poor treatment outcomes according to clinical and demographic characteristics and social risk factors (SRFs) in a 2013–18 cohort. 25 semi-structured interviews were undertaken in purposely selected individuals (9 patients and 16 healthcare professionals) to glean insights on their healthcare priorities and the factors that contribute to poor treatment outcome.

### Results

The quantitative cohort comprised 2252 patients. Those who were ≥ 55 years of age, foreign-born from Central Europe, East Asia and Sub Saharan Africa and with MDR-TB were more likely to have poor treatment outcomes. According to patients and healthcare professionals, the factors that contribute to vulnerability to develop TB and poor treatment outcomes include poor working and living conditions, inadequate or absent welfare protection,

**Funding:** The research was supported by the National Institute for Health and Care Research (NIHR) Health Protection Research Unit (HPRU) in Respiratory Infections at Imperial College London, UK, in partnership with the UK Health Security Agency (London, UK), NIHR200927. The views expressed are those of the author(s) and not necessarily those of the National Health Service, the NIHR, the Department of Health and Social Care, or the UK Health Security Agency. MP is supported by UKRI/MRC/NIHR (MR/V027549/1).

**Competing interests:** The authors have declared that no competing interests exist.

poor primary healthcare responsiveness, treatment duration and side effects. These factors could be addressed by increased networking, partnership and integration between healthcare and social services and better integration between primary and secondary healthcare.

## Conclusions

In both cities, being ≥ 55 years of age, having MDR-TB and being of foreign-birth are predictors of unfavourable treatment outcome. Risk of poor treatment outcome and vulnerability seem to be multidimensional. A better understanding of specific vulnerabilities and how they affect patient care pathway is needed to design adequate support programmes.

## Introduction

Tuberculosis (TB) re-emerged as a public health problem in England in the late 1990s, reaching peak incidence in 2011, with considerable decline thereafter: England recorded its lowest ever TB incidence rate (8.3 per 100,000 population) in 2018, and although a rise of 2.4% was reported in 2019, with a rate of 7.3 per 100,00 population in 2020, England's rate is below the WHO definition of a low incidence country [1]. Yet, this decline heightens challenges for TB prevention and care, because epidemiological heterogeneity will increase [2]. There is higher incidence of TB amongst the most deprived sections of the population, who have specific risk factors linked to marginalization e.g., homelessness or alcoholism [1,3]. In addition, as overall TB rates decline, the proportion requiring enhanced care will increase [4].

In the context of TB prevention and care, the term 'underserved' denotes heterogeneous subgroups of the population that due to their social circumstances or lifestyle have difficulty accessing TB care and treatment and experience diagnostic delays that lead to poor treatment adherence and poor treatment outcomes [1,4]. In England, underserved populations are identified by the presence of predefined social risk factors (SRFs) (i.e., drug or alcohol problems, homelessness, imprisonment) [4]. One goal of the Collaborative Tuberculosis Strategy for England 2015–20 was to tackle TB in underserved populations; this required bespoke healthcare service models [5]. The Strategy encouraged novel outreach interventions [5], but systematic reviews demonstrate that intervention efficacy is setting-specific [6]; the National Institute for Health and Care Excellence (NICE) recommends that intervention appropriateness should be informed by a local needs assessment [6]. For this, the target population must be first defined and characterised to enable the identification of their healthcare priorities [6]. Nevertheless, individuals and groups are not homogeneous and, therefore, universal assumptions about a group based on the presence of specific SRFs could mask diverse sources of interrelated vulnerabilities e.g., income distribution or lack of social capital [7]. These vulnerabilities are driven by a multidimensional spectrum of factors and structural barriers [7,8].

Most studies on TB treatment outcomes in underserved populations in England have been done in London [9,10], however this may not be representative of the national picture. A knowledge gap exists for the health priorities of underserved populations in other urban settings with a high TB burden [1]. This is relevant for informing setting-specific policy making. We therefore performed an epidemiological characterization of TB patients from Birmingham and Leicester—two cities in the Midlands with the highest numbers of TB cases outside London with TB rates of 9.2 and 6.4 per 100,000 respectively, and similar epidemiological patterns; the disease affects mainly non-UK born individuals [1]—to identify those more likely to have poor treatment outcomes, and to glean insights from patients and their healthcare providers,

on their healthcare priorities, barriers for healthcare access, additional services and interventions needed to improve healthcare delivery and the factors that contribute to poor treatment outcome.

## Materials and methods

### Study design, setting and study participants

We performed an equal status mixed methods study with concurrent design [11] in two National Health Service (NHS) trusts across Birmingham and Leicester that combined retrospective epidemiological data and prospectively collected qualitative data. Quantitative data sources included local TB registers and the national Enhanced TB Surveillance (ETS) system including all incident TB cases $\geq$ 18 years of age reported in Birmingham and Leicester from 2013 to 2018.

The key informants for the qualitative study were purposively selected among TB patients and healthcare professionals. The local lead nurses and consultants supported the identification of TB patients from diverse backgrounds, and made the initial approach explaining the purpose of the study and inviting participation in a semi-structured interview. Individuals were eligible to participate in the qualitative study if they were $\geq$ 18 years of age, had a TB diagnosis and experienced diagnostic delay of more than two months from the first appearance of symptoms to the initiation of treatment or poor treatment outcome, and could provide informed consent. Healthcare professionals currently working at one of the two study sites with experience in care and management of TB patients were also eligible. Patients with no data available on final treatment outcome were excluded because it was the primary outcome for quantitative analysis.

All participants in the qualitative study were accessed at the most appropriate and convenient place after discussion and agreement between them and the local lead TB nurse who first approached them and inquired about their willingness to participate in the study. The local lead TB nurse assessed participant's capacity, and asked for written consent to participate in the study after a full explanation was given, an information leaflet offered and time allowed for consideration. Written participant consent was obtained, the consent included permission for audio recording of their interviews with subsequent written transcription. TB patients received a £20 reimbursement to cover travel expenses. Interviews with healthcare professionals were done by the investigators LCBA and NK, while HP interviewed TB patients. None of the interviewers had previous relationships or interactions with the interviewees that could have affected the interview process or interpretation of results. Similar topic guides were used for interviewing patients and health care professionals Tables 1 and 2, respectively. The topic guides were developed based on discussions with healthcare providers and informed by the goals of the Collaborative Tuberculosis Strategy for England 2015–20 [5]. Transcripts were quality assured by AB.

The interviews with healthcare professionals were performed as part of a service evaluation and development initiative by the UK Health Security Agency (UKHSA). The study was given a favourable ethical opinion by the London Central Research Ethics Committee and was granted ethical approval by the Health Research Authority (reference 16/LO/2051).

### Data analysis and integration

**Quantitative analysis.** The primary outcome for quantitative analysis was poor treatment outcome defined as a composite of any or all of; non-adherence to treatment (including those due to side effects), lost to follow-up or death (TB cause of death, contributed to death, incidental to death and relationship between TB and death unknown) according to the definitions used by the UKHSA [1]. Sociodemographic and clinical characteristics of study participants

**Table 1. Topic guide for interviewing TB patients.**

| No. | Question |
|---|---|
| 1 | Introduction |
| 2 | Would you be able to talk me through your patient experience? Starting from when you first experienced symptoms of TB, how you were diagnosed, your treatment and whether you are currently taking treatment or have completed treatment. |
| 3 | How long did you experience TB symptoms before you tried to seek medical attention? |
| 4 | Was there anything that prevented you from seeking medical care or made it difficult to access healthcare? |
| 5 | Are there any other barriers for accessing healthcare? |
| 6 | Do you think there are any patients who are underserved by the TB services? |
| 7 | What do you think should be added to the TB services? |
| 8 | Do you think community based interventions such as raising awareness of TB, TB treatment and healthcare services available would encourage people to access healthcare? |
| 9 | What do you think are the biggest challenges patients face when trying to complete TB treatment? |
| 10 | Did you receive any services which helped you to complete treatment? |
| 11 | What should be added to the TB service to help patients complete treatment? |
| 12 | Do you think an establishment for TB patients which provides accommodation, nursing support and social support would be useful? |
| 13 | Do you feel any further services need to be provided after TB treatment is completed? |

are depicted as total counts and percentages. We used multiple imputation by chained equations [12] to produce imputed values for missing quantitative variables; 20 imputed data sets were created and analysed according to Rubin's rules [13]. A detailed description of the percentage of missing information for each variable and the imputation method is given in the supporting information and S1 Table in S1 File. The risk factors included in the analysis have been reported to be significantly associated with poor outcome [9,10]. To identify their association with our primary outcome, each imputed dataset was analysed separately using univariate and multivariate logistic regression models, and the results were combined into a single multiple-imputation result [13]. The results are presented as odds ratios (OR) with 95% CIs and two-sided p values. We did a sensitivity analysis to account for the imputation method using complete case analysis S3 Table in S1 File. Stata version 15.1 was used for all statistical analysis.

**Table 2. Topic guide for interviewing health care providers.**

| No. | Question |
|---|---|
| 1 | Introduction: Could you please describe your role in the TB clinic |
| 2 | What does the term underserved mean to you? |
| 3 | How would you relate that term to patients with TB? |
| 4 | There are several sub-populations that are considered being underserved in terms of health services: persons experiencing homelessness, persons with drug or alcohol problems, individuals non-UK born, persons who have been in prison. Would you say that most of your patients belong to one of these groups or how often do you meet people from these group in your daily practice? |
| 5 | Do the patients with TB in Birmingham/Leicester have any special socio-demographic characteristic that you could highlight? |
| 6 | Which type of patients do you consider underserved and what extra care do they require? |
| 7 | Are there any barriers for engaging with these patients? |
| 8 | What do you think underserved patients see as the main barriers to their access to health care? |
| 9 | What type of special service do you provide for underserved groups? |
| 10 | What do you think should be added to these services? |
| 11 | According to your experience, what characteristics an ideal service for underserved populations should have? |

**Qualitative analysis.** Anonymised verbatim transcripts of the semi-structured interviews were analysed by ML and JB using thematic analysis and adopting an interpretivist approach. The process of developing codes and themes was both inductive (data-driven) and deductive (theoretical) [14,15]. The Social Determinants of Health framework was used as the theoretical foundation, based on its assertion that social exclusion is a foundational cause of health inequity [16,17]. Data familiarisation was the first analytical step. Each transcript was read twice and a record of initial impressions was kept. Afterwards, *NVivo 11* software was used to code each transcript, to merge and rearrange codes and to create candidate themes. Word frequency and text search tools were used to explore the coded data. A draft mind-map was developed to visualise candidate themes and codes, and the concept of thematic networks was used to organise the data [18]. Data saturation was achieved for all themes presented in the results section. Additional information regarding the consolidated criteria for reporting qualitative research (COREQ) for this study is given in the supporting information.

**Data integration.** The qualitative and quantitative data were collected in parallel, analysed separately and then integrated through narrative merging using a weaving style, whereby quantitative and qualitative findings are presented on a theme-by-theme basis [19].

## Results

The study cohort for the quantitative analysis comprised 2252 patients; their sociodemographic and clinical characteristics are depicted in Table 3. This epidemiological characterization showed that in Birmingham and Leicester, TB affects young socioeconomically active people, predominantly male (59%) who live in the most deprived areas (62%) Table 3. The presence of SRFs was low and varied across the two cities. 1,043 (46.4%) patients had pulmonary TB, 21 (1.1%) had MDR-TB and 1,996 (88.6%) completed treatment Table 3.

We conducted 25 semi-structured interviews, 9 with patients diagnosed with pulmonary TB and 16 with healthcare professionals; their sociodemographic characteristics are shown in Table 4. Three main themes emerged from the qualitative analysis: two themes on barriers and factors related to social aspects of TB and healthcare provision, and a third on solutions to address these barriers Fig 1.

The epidemiological analysis of factors associated in the multivariate analysis with poor treatment outcome demonstrated that in both settings patients who are $\geq 55$ years of age and foreign-born individuals from Central Europe, East Asia and Sub Saharan Africa are more likely to have poor treatment outcomes than patients of younger age or UK-born (odds ratio (95% CI); 2.0 (1.2–3.5), 6.1 (3.1–12), 4.2 (1.2–14) and 1.6 (1.0–2.7), respectively Table 5. There was no significant association of poor outcomes either with index of deprivation or any of the SRFs assessed Table 5. However, patients with MDR-TB were more likely to have poor treatment outcomes Table 5. Results of the univariate analysis are shown in S2 Table in S1 File. All associations remained significant after accounting for missing data with a sensitivity analysis S3 Table in S1 File. Drawing on the quantitative findings, the qualitative results explore patients' diagnostic and treatment journey, other factors contributing to poor treatment outcomes, barriers for healthcare access, challenges for treatment completion, and additional services and interventions needed to improve healthcare delivery and treatment completion Tables 1 and 2. The qualitative results for both cities are presented together because we did not find differences that merit their separation.

### Social aspects of TB

The themes on barriers and factors related to social aspects of TB and healthcare provision highlight the complexity of TB and the intertwined issues that act at individual- and

**Table 3. Baseline characteristics for all study participants and across cities.**

| | Birmingham (n = 1486) (%) | Leicester (n = 766) (%) | All participants (n = 2252) (%) |
|---|---|---|---|
| **Age (years)** | | | |
| 16–24 | 153 (10.3) | 120 (15.6) | 273 (12.1) |
| 25–34 | 342 (23) | 176 (22.9) | 518 (23) |
| 35–44 | 320 (21.5) | 171 (22.3) | 491 (21.8) |
| 45–54 | 226 (15.2) | 112 (14.6) | 338 (15) |
| 55–65 | 195 (13.1) | 90 (11.7) | 285 (12.6) |
| >65 | 250 (16.8) | 97 (12.6) | 347 (15.4) |
| **Sex** | | | |
| Male | 898 (60.5) | 426 (55.6) | 1324 (58.8) |
| **Region of origin** | | | |
| Americas | 25 (1.7) | 1 (0.1) | 26 (1.1) |
| Central Europe | 47 (3.2) | 16 (2.1) | 63 (2.8) |
| East Asia | 10 (0.6) | 5 (0.6) | 15 (0.6) |
| East Europe | 5 (0.3) | 3 (0.4) | 8 (0.3) |
| East Mediterranean | 12 (0.8) | 2 (0.2) | 14 (0.6) |
| North Africa | 26 (1.7) | 10 (1.3) | 36 (1.6) |
| South Asia | 707 (48.1) | 428 (58.5) | 1135 (51.5) |
| South East Asia | 21 (1.4) | 16 (2.1) | 37 (1.6) |
| Sub Saharan Africa | 215 (14.6) | 109 (14.9) | 324 (14.7) |
| United Kingdom | 372 (25.3) | 129 (17.6) | 501 (22.7) |
| West Europe | 30 (2) | 12 (1.6) | 42 (1.9) |
| **Deprivation index** | | | |
| 1–2 decile (most deprived) | 1089 (73.6) | 269 (37.7) | 1358 (61.9) |
| 3–4 decile | 181 (12.2) | 262 (36.7) | 443 (20.2) |
| 5–6 decile | 116 (7.8) | 83 (11.6) | 199 (9) |
| 7–8 decile | 57 (3.8) | 55 (7.7) | 112 (5.1) |
| 9–10 decile (least deprived) | 36 (2.4) | 44 (6.1) | 80 (3.6) |
| **Drug problems** | | | |
| Yes | 70 (4.7) | 6 (0.9) | 76 (3.5) |
| **Homelessness** | | | |
| Yes | 33 (2.2) | – | 33 (2.2) |
| **Imprisonment** | | | |
| Yes | 50 (3.3) | – | 50 (3.3) |
| **Alcohol problems** | | | |
| Yes | 27 (1.8) | 32 (4.8) | 59 (2.7) |
| **Disease site** | | | |
| Pulmonary | 795 (53.7) | 248 (32.3) | 1043 (46.4) |
| Extra-pulmonary | 552 (37.3) | 354 (46.2) | 906 (40.3) |
| Both | 132 (8.9) | 164 (21.4) | 296 (13.1) |
| **MDR-TB** | | | |
| Yes | 19 (1.2) | 2 (0.6) | 21 (1.1) |
| **Treatment outcomes** | | | |
| Died | 78 (5.2) | 33 (4.3) | 111 (4.9) |
| Lost to follow up | 56 (3.8) | 45 (5.9) | 101 (4.5) |
| Treatment stopped | 28 (1.9) | 16 (2.0) | 44 (1.9) |
| Treatment completed | 1324 (89.1) | 672 (87.7) | 1996 (88.6) |

*(Continued)*

**Table 3.** (Continued)

| | Birmingham (n = 1486) (%) | Leicester (n = 766) (%) | All participants (n = 2252) (%) |
|---|---|---|---|
| **Overall outcome** | | | |
| Completed | 1324 (89.1) | 672 (87.7) | 1996 (88.6) |
| Poor outcome | 162 (10.9) | 94 (12.2) | 256 (11.3) |

community-levels, creating vulnerability and obstacles along the patient care pathway. Structural barriers surrounding work and access to welfare support create situations where people would be left without money to live on, through work, or through receiving welfare benefits.

*'Many people we see are often in low paid jobs that don't get sick pay if they are off, so if they're infectious, then they shouldn't be going to work but where do they get that money from? That's the real difficulty sometimes.' (TB-Nurse 3, Female)*

*'When we applied for benefits, we couldn't get benefits, because I have TB, I couldn't go to employment centre because I had TB.' (Patient 6, Male)*

Some TB patients have complex lives with issues such as immigration status, access to housing, and co-morbidities, which challenge their abilities to address TB.

*'We do spend a lot of time trying to explain to people we are not the police, we are not going to report you to anyone, but I doubt they believe us.' (TB consultant 1, Male)*

*'Some have been in and out of prison and as a consequence end up with problems with their housing, or problems with mental health, or problems with drugs. They do seem to be very much linked.' (Nurse 2, Female)*

The groups of patients that healthcare professionals consider especially vulnerable are those with drugs and alcohol use disorder, elderly patients, low-paid workers, those with mental health disorders, with chaotic lifestyles, under prolonged stress and some foreign-born individuals.

**Table 4. Sociodemographic characteristics of participants in semi-structured interviews.**

| Health care providers (n = 16) | No. | Patients (n = 9) | No. |
|---|---|---|---|
| **Occupation** | | **Country of origin** | |
| Chest physician | 5 | UK | 4 |
| TB nurse | 5 | India | 2 |
| Lead TB nurse | 2 | Bangladesh | 1 |
| Nurse | 2 | Eritrea | 1 |
| Support worker | 1 | Lithuania | 1 |
| Cultural link worker | 1 | | |
| **Sex** | | **Sex** | |
| Female | 10 | Female | 5 |
| Male | 6 | Male | 4 |
| **Workplace** | | **Treatment location and residency** | |
| Birmingham | 7 | Birmingham | 3 |
| Leicester | 9 | Leicester | 6 |

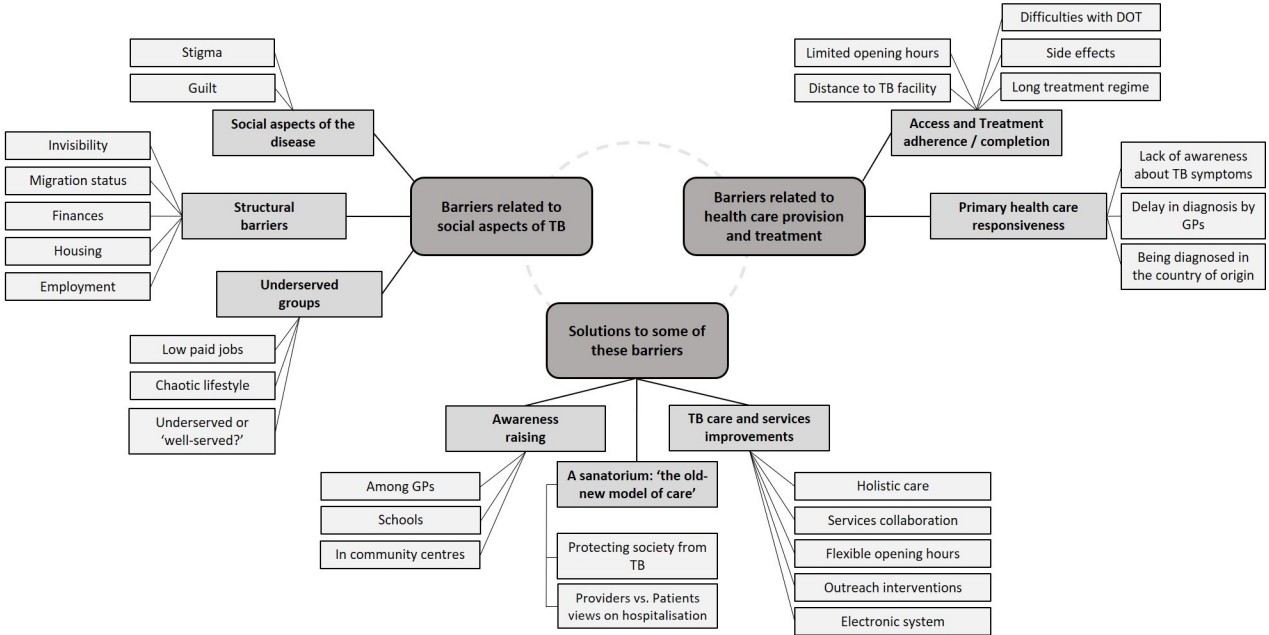

**Fig 1. Main themes and codes derived from the participants' perspectives.** GP general practitioner, DOT direct observation of therapy.

*'You have a very elderly frail old person at home and it's difficult to get these people to adhere to their medication because they just don't have the support in place.' (TB-Nurse 2, Female)*

*'People with drug and alcohol problems who perhaps had bad experiences in life generally and find it difficult to trust any authority.' (Support worker, Female)*

## Healthcare provision for TB

Among some healthcare professionals the term 'underserved' did not adequately reflect the range of services available and the effort and commitment that healthcare professionals devote to engaging and supporting patients.

*'People who are generally not compliant who just need a lot more support. But that doesn't necessarily mean they are underserved. We are giving them the support, they are just not engaging with us.' (Lead TB nurse 1, Female)*

*'I wouldn't say they are underserved though because they've got so many services that they can access.' (TB-Nurse 1, Female)*

However, both patients and healthcare professionals agreed that the time and location of services represent barriers for access, and once the services are reached, there is often poor primary healthcare responsiveness.

*'Most of them are working, they don't get the right appointment at the right time on day offs.' (Patient 3, Female)*

*'Clinics are held in the edge of the city rather than in the middle and if patients are all in the middle of the city without transport, without money, you know, that could be difficult for them. (Support worker, Female)*

**Table 5. Multivariate analysis of factors associated with poor treatment outcome.**

| | Birmingham OR (95% CI) | p-value | Leicester OR (95% CI) | p-value | All participants OR (95% CI) | p-value |
|---|---|---|---|---|---|---|
| **Age (years)** | | | | | | |
| 16–24 | 1.2 (0.5–2.8) | 0.668 | 2.4 (0.7–7.6) | 0.122 | 1.5 (0.8–2.7) | 0.112 |
| 25–34 | Ref | | Ref | | Ref | |
| 35–44 | 1.1 (0.5–2.2) | 0.718 | 1.3 (0.4–4.1) | 0.608 | 1.1 (0.6–1.8) | 0.649 |
| 45–54 | 1.8 (0.9–3.6) | 0.093 | 1.0 (0.2–3.9) | 0.984 | 1.4 (0.8–2.5) | 0.142 |
| 55–65 | 2.3 (1.1–4.8) | 0.014 | 1.1 (0.3–4.5) | 0.791 | 2.0 (1.2–3.5) | 0.007 |
| >65 | 8.6 (4.6–16) | <0.0001 | 4.4 (1.3–14) | 0.012 | 7.3 (4.6–11) | <0.0001 |
| **Sex** | | | | | | |
| Female | Ref | | Ref | | Ref | |
| Male | 1.3 (0.9–2.0) | 0.106 | 1.1 (0.5–2.1) | 0.774 | 1.2 (0.9–1.6) | 0.130 |
| **World region of origin** | | | | | | |
| Americas | 1.8 (0.6–5.7) | 0.283 | – | – | 2.3 (0.8–6.4) | 0.110 |
| Central Europe | 4.8 (2.0–11) | <0.0001 | 32 (5.2–200) | <0.0001 | 6.1 (3.1–12) | <0.0001 |
| East Asia | 0.9 (0.1–8.4) | 0.947 | 34 (1.6–106) | 0.042 | 4.2 (1.2–14) | 0.020 |
| East Europe | 3.5 (0.3–36) | 0.293 | 14 (0.8–223) | 0.060 | 4.5 (0.8–25) | 0.080 |
| East Mediterranean | 2.4 (0.4–12) | 0.300 | – | – | 3.2 (0.8–12) | 0.095 |
| North Africa | – | – | – | – | – | – |
| South Asia | 1.0 (0.6–1.7) | 0.800 | 1.2 (0.4–3.7) | 0.698 | 1.2 (0.8–1.8) | 0.272 |
| South East Asia | 1.5 (0.3–6.2) | 0.549 | – | – | 1.0 (0.2–3.8) | 0.948 |
| Sub Saharan Africa | 1.4 (0.7–2.6) | 0.315 | 2.2 (0.6–7.3) | 0.183 | 1.6 (1.0–2.7) | 0.039 |
| United Kingdom | Ref | | Ref | | Ref | |
| West Europe | 0.7 (0.1–2.7) | 0.638 | 3.8 (0.6–22) | 0.135 | 1.2 (0.4–3.4) | 0.603 |
| **Deprivation index** | | | | | | |
| 1–2 decile (most deprived) | Ref | | Ref | | Ref | |
| 3–4 decile | 1.0 (0.6–1.8) | 0.881 | 0.5 (0.2–1.0) | 0.071 | 1.1 (0.7–1.5) | 0.595 |
| 5–6 decile | 0.7 (0.3–1.6) | 0.530 | 0.1 (0.01–0.9) | 0.042 | 0.6 (0.3–1.1) | 0.104 |
| 7–8 decile | 1.3 (0.5–3.1) | 0.467 | 0.1 (0.01–1) | 0.060 | 0.7 (0.3–1.5) | 0.461 |
| 9–10 decile (least deprived) | 0.6 (0.1–2.4) | 0.575 | 0.3 (0.03–2.5) | 0.275 | 0.7 (0.3–1.6) | 0.497 |
| **Drug problems** | | | | | | |
| Yes | 1.5 (0.5–4.3) | 0.371 | – | – | 1.4 (0.5–3.6) | 0.403 |
| **Homelessness** | | | | | | |
| Yes | 1.1 (0.2–4.9) | 0.845 | – | – | 1.1 (0.2–4.2) | 0.879 |
| **Imprisonment** | | | | | | |
| Yes | 1.2 (0.4–3.6) | 0.674 | – | – | 1.5 (0.5–4.0) | 0.415 |
| **Alcohol problems** | | | | | | |
| Yes | 2.0 (0.6–6.8) | 0.247 | 1.6 (0.4–6.2) | 0.444 | 1.8 (0.6–4.9) | 0.223 |
| **Disease site** | | | | | | |
| Pulmonary | Ref | | Ref | | Ref | |
| Extra-pulmonary | 1.2 (0.8–1.8) | 0.326 | 0.6 (0.2–1.3) | 0.234 | 1.0 (0.6–1.5) | 0.989 |
| Both | 1.5 (0.8–2.8) | 0.184 | 0.5 (0.2–1.4) | 0.252 | 1.0 (0.7–1.3) | 0.954 |
| **MDR-TB** | | | | | | |
| Yes | 5.3 (1.7–15) | 0.003 | – | – | 3.4 (1.2–9.8) | 0.019 |

Within primary care, TB is not a diagnosis general practitioners (GPs) immediately consider when patients arrive with symptoms consistent with pulmonary TB. This leads to diagnostic delays and progression of the disease prior to treatment onset.

*'I was having chest infections and getting antibiotics, after about a week it was getting bad again. Then my brother in law found out that he had got TB and he says, "if I was you I would just get checked out, just to be on the safe side". I went to my GP, and he said I couldn't have TB, because we don't have it in where I live, it's like a country town.' (Patient 6, Male)*

*'I go to the GP, the first time he gave me nasal spray, I think he thinks something else not TB, and one month I go again so he gave a blood test, some blood test and he says it's nothing, he gave vitamin D only.' (Patient 7, Male)*

Once diagnosed, contributing factors to poor treatment adherence and completion were the treatment duration, side effects, amount of tablets and difficulties to cope with the direct observation of therapy (DOT) programme.

*'It was like 24 tablets or something like that in one go. It was a lot, they are not small tablets; some of them are massive. So that, I struggled with that, I had to chop them up. It does make you feel sick just drinking water and tablets.' (Patient 8, Female)*

*'My liver got affected and they sent me for a test up here and they said my liver deteriorated, taking all these tablets so I had to come off them.' (Patient 6, Male)*

## Potential solutions to address these barriers

Some recommendations to improve existing services consist of health promotion having a wider focus, not exclusive to TB, and better networking between different services, including social support and social workers within the TB team.

*'I think it's really good, having a worker that wasn't necessarily designated within the role of TB to go out and help our TB patients with things like benefits, housing applications.' (TB nurse 2, Female)*

*'Better collaboration and communication with social services, with the housing teams, the council in general. If we had a better network of communication with them and a more organised way of connecting with the voluntary sector, I think that would help a lot.' (Nurse 2, Female)*

In addition to connecting diverse services, supplemental services through drop-in clinics and community and peer advocates were recognised by patients and healthcare professionals.

*'We could have more drop-in services, places people can just pop in and say "I'm just wondering about this" so that you have a greater scope of capture of people who happen to be in a particular place.' (TB consultant 5, Male)*

*'The best thing would be working with community centres, because that is a place where I am not seeing people purely talking about infectious diseases.' (Patient 4, Female)*

Moreover, connectivity of medical records through technology would also make for improved healthcare access and provision.

*'The fact is they write to your GP but there was nothing there, nothing was on the computer, and I'm like, "don't you guys talk to each other?" Shouldn't it be once someone has some problem, shouldn't it be updated on your records?' (Patient 5, Female)*

*'There's a lot more that could be done if we had an integrated IT system for primary and secondary care. I think there's a huge opportunity missed here across the board for making health service as a whole more cost-effective.' (TB consultant 3, Male)*

When reflecting on best models of care, healthcare professionals and patients shared divergent views, with practitioners wishing for more control over patients and patients wanting to maintain a sense of normal life.

*'I think we need somewhere we can put patients who don't have good housing or who just have a really complex lifestyle to try and take them out of that lifestyle and where they can get the services they need and access what they need.' (TB nurse 1, Female)*

*'I would prefer to be at home than in hospital, but I suppose if you're homeless or whatever then I suppose that would be quite good, but for me I was better off at home really.' (Patient 6, Male)*

## Discussion

This health needs assessment demonstrates that in Birmingham and Leicester, being ≥ 55 years of age, of foreign-birth or having MDR-TB are significant independent factors associated with poor treatment outcome. According to patients and healthcare professionals, the factors that contribute to develop TB and poor treatment outcomes include working and living conditions, inadequate or absence of welfare protection, poor primary healthcare responsiveness, treatment duration and side effects. These factors could be addressed by increased networking, partnership and integration between healthcare and social services, better integration between primary and secondary healthcare and easily accessible community-based services.

Vulnerability is defined as the lack of means to cope with external risks, contingencies, stress or shocks that results in personal harm or damaging loss [20]. Vulnerability is multidimensional, and encompasses individual abilities and actions, the readiness of instrumental support, and community resources that may ease or thwart personal response [21]. In the context of TB, it could be considered twofold: vulnerability to developing TB because of exposure to internal and external factors and, thereafter, vulnerability to poor treatment outcome. In England, in line with our findings, the proportion of patients with poor treatment outcome is higher in those born outside the UK, over 65 years of age, and with MDR-TB [1]. However, in contrast to our results, a higher percentage of patients with SRFs have poor treatment outcome compared to those without SRFs [1]. The association of SRFs and poor treatment outcome in TB patients has been well described in London [9,10], leading to the implementation of effective interventions [22]. The difference in treatment outcomes in patients with SRFs, according to geographical area, is likely explained by the fact that the national estimates use aggregate data which cannot uncover geographical variation [1]. TB epidemiology is heterogeneous [2], and the proportion of patients with SRFs in these cities was very small. Thus, although we found no relationship between SRFs and poor outcomes, we cannot exclude the possibility of an association. In our study, among foreign-born patients, South Asians form the highest proportion but do not experience poorer outcomes. We hypothesise that this may be due to their adaption, over several decades, to the healthcare system compared with individuals recently arrived from other regions of the world. In addition, these results could also be due to the

effectiveness of bespoke strategies devised and deployed by teams of healthcare and public health professionals cognisant of local epidemiology and patients' needs [4].

In our patients' cohort, apart from poor primary healthcare responsiveness, other factors related to poor treatment outcomes mentioned by healthcare professionals and TB patients are beyond what healthcare services alone can address. The long treatment length and its toxicity are inherent to current TB therapeutics [23]; the presence of structural factors is due to policies aimed to restrict welfare and healthcare access for foreign-born individuals [24–26]; and to reforms to the welfare system in general [27]. Thus, while a shorter and less toxic treatment regimen is a long-awaited breakthrough for global TB prevention and care [23], the spectrum of vulnerabilities arising from the social and economic ecosystem are best addressed by social protection and welfare policy.

Given that the most pressing factors that influence TB treatment outcomes are not address-able by the healthcare sector alone, it is unsurprising that patients and healthcare professionals long for increased networking, partnership and integration between healthcare and social ser-vices, along with better integration between primary and secondary healthcare and easily accessible community-based services. Health outcomes are affected by the combined effect of public health and social care policies [28]: effective social care can prevent people from need-ing hospitalisation and can expedite discharge once hospitalised, this is of particular impor-tance for older patients. In England, social care is the remit of local authorities and not the NHS, and although there has been great emphasises on the need for more integration of care [29], there is little evidence on effective implementation [30]. On the other hand, nationwide attempts to integrate NHS primary and secondary care IT systems have been slow and trouble-some [31].

The policies that restrict healthcare access and welfare protection for foreign-born individu-als create hard-to-surmount barriers [26,32], which preclude protection from the factors that enhance vulnerability to poor treatment outcome. It is possible to prevent the development of active TB in foreign-born individuals, through screening and treatment of latent TB infection, an intervention proven to be effective [33,34], though the participation in this programme is impeded by the same structural factors [35]. We showed previously that some of these barriers could be sidestepped by increasing collaboration with civil society organizations to develop and implement community-based services [35].

This study has some limitations. In Leicester, the local TB register did not contain data on homelessness or imprisonment and our results could be affected by this missing information. However, we accounted for this source of uncertainty and bias by, first using multiple imputa-tion and sensitivity analysis which showed no difference between the imputed and the com-plete case analysis; and second, by obtaining insights from patients and healthcare professional through semi-structured interviews which showed that in these two cities these are not the most pressing issues. Moreover, we did not have data available on the presence of co-morbidi-ties that may influence treatment outcomes. We identified MDR-TB as a factor contributing to poor treatment outcomes; however, these patients are well-recognised as vulnerable to worse outcome due to the ineffectual and prolonged treatment. Furthermore, all patients interviewed were diagnosed with pulmonary TB, therefore, no insights from patients with extrapulmonary TB were included; they can face different challenges not covered in this study. For this reason, we could not address the complexities around diagnosing extrapulmonary TB in contrast to pulmonary TB in a primary care setting. Likewise, some of the most vulnerable individuals could be less likely to have had their TB diagnosed and thus been included in this study. Cap-turing or assessing social vulnerability is difficult to do [36], and we were constrained by the approach used by the UKHSA of measuring only certain SRFs. Vulnerability is not explained by one social or demographic factor alone; it is influenced by multiple and multidimensional

factors. We attempted to overcome this limitation by integrating qualitative and quantitative data to comprehend patients' and service providers' perspectives on the hurdles for getting a diagnosis and completing treatment. There may be biases to the sampling strategy which was led by TB nurses given their specific role in this context. Some patients declined the invitation to participate in the study, thus, our results may be biased towards the views of those who opted to be interviewed. However, although the patient sample in the qualitative component of the study was small, reaching the vulnerable is itself a challenge, and using a mixed methods methodology allowed us to include voices often unheard in policy or research.

Our study suggests that there is a need for multisectoral partnership and political commitment to address and reduce vulnerability to poor TB outcomes and reduce inequities within the urban context. Local authority public health departments are in the best position to coordinate the care of their most vulnerable residents as they oversight access to housing, welfare and social care. They could work in partnership with the NHS and aim to facilitate integration of health and welfare system to provide support (e.g., financial, housing, mental health and immigration advice), regardless of immigration status, to patients and their families.

In conclusion, heterogeneity of TB epidemiology across England is associated with loco-regional differences in treatment outcomes across populations and settings. The national estimates must be analysed using disaggregated data to inform appropriate context-specific prevention and care policies. Risk to poor TB treatment outcome and vulnerability are multidimensional and cannot be reduced to a single sociodemographic characteristic. A better understanding of specific vulnerabilities, how they affect TB treatment outcomes, and their relation to structural barriers outside the scope of individuals and healthcare providers is needed to design adequate support programmes. This should be region-specific and informed by local TB networks and prevention and care boards.

## Supporting information

**S1 File. Contains supporting tables.**
(DOCX)

## Acknowledgments

The authors thank all TB patients and healthcare providers who participated in the study.

## Author Contributions

**Conceptualization:** Luis C. Berrocal-Almanza, Onn Min Kon, Dominik Zenner, Ajit Lalvani.

**Data curation:** Luis C. Berrocal-Almanza, Helen Piotrowski, Julie Botticello, Nisha Karnani, Hanna Kaur.

**Formal analysis:** Luis C. Berrocal-Almanza, Marcela Lima, Helen Piotrowski, Julie Botticello, Amarjit Badhan, Nisha Karnani.

**Funding acquisition:** Dominik Zenner, Ajit Lalvani.

**Investigation:** Luis C. Berrocal-Almanza, Hanna Kaur, Manish Pareek, Pranabashis Haldar, Martin Dedicoat, Onn Min Kon.

**Methodology:** Luis C. Berrocal-Almanza, Onn Min Kon.

**Project administration:** Luis C. Berrocal-Almanza, Helen Piotrowski, Amarjit Badhan.

**Resources:** Hanna Kaur, Pranabashis Haldar, Martin Dedicoat.

**Supervision:** Onn Min Kon, Dominik Zenner.

**Writing – original draft:** Luis C. Berrocal-Almanza.

**Writing – review & editing:** Luis C. Berrocal-Almanza, Marcela Lima, Helen Piotrowski, Julie Botticello, Amarjit Badhan, Nisha Karnani, Hanna Kaur, Manish Pareek, Pranabashis Haldar, Martin Dedicoat, Onn Min Kon, Dominik Zenner, Ajit Lalvani.

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
