## [Decision Letter · Decision Letter 0]

17 Nov 2022

PONE-D-22-28286Vulnerability and tuberculosis treatment outcomes in urban settings in England: a mixed-methods study.PLOS ONE

Dear Dr. Berrocal-Almanza,

Thank you for submitting your manuscript to PLOS ONE. After careful consideration, we feel that it has merit but does not fully meet PLOS ONE’s publication criteria as it currently stands. Therefore, we invite you to submit a revised version of the manuscript that addresses the points raised during the review process.

We look forward to receiving your revised manuscript.

Kind regards,

Tom Wingfield

Academic Editor

PLOS ONE

Journal Requirements:

" ext-link-type="uri" xlink:type="simple">https://journals.plos.org/plosone/s/file?id=ba62/PLOSOne_formatting_sample_title_authors_affiliations.pdf"

"The research was funded by the National Institute for Health Research Health Protection Research Unit (NIHR HPRU) in Respiratory Infections at Imperial College London (London, UK) in partnership with the UK Health Security Agency (London, UK). The views expressed are those of the author(s) and not necessarily those of the National Health Service, the NIHR, the Department of Health, or the UK Health Security Agency. AL was supported by the NIHR Imperial Biomedical Research Centre. MP is supported by a NIHR Development and Skills Enhancement Award and UKRI/MRC/NIHR (MR/V027549/1)."

"AL was supported by the NIHR Imperial Biomedical Research Centre. MP is supported by a NIHR Development and Skills Enhancement Award and UKRI/MRC/NIHR (MR/V027549/1). The funders had no role in study design, data collection and analysis, decision to publish, or preparation of the manuscript."

Additional Editor Comments:

The research conducted is important within the context of aiming for TB elimination in the UK and considering strategies that reach underserved groups in all areas of the country. The reviewers raised a number of concerns about the manuscript, which I have tried to summarise below.

Despite one of the reviewers rejecting the manuscript, I have decided to recommend the manuscript for resubmission following major revisions. Following resubmission, I will then ask the reviewers for their further review and recommendations.

In summary, the reviewers commented that:

- It would be helpful to have further explanation of the local epidemiology of TB in Birmingham and Leicester and to reflect on the difference between TB epidemiology (and indeed cohorts) in the interpretation of results

- One reviewer reported that the manuscript read like separate studies (and a service evaluation) which had been brought together post hoc for the purposes of analysis and left the discussion section struggling to "weave" together a comprehensive narrative the qualitative and quantitative results.

- Qualitative methods would benefit from more details on: the reasons behind the use of mixed methods (including whether the study was originally intended to be mixed methods); the selection of participants including reason for diagnostic delay in inclusion criteria (plus expansion of discussion section with relation to diagnostic delay); how topic guides were developed; and the theoretical frameworks used to inform the qualitative analysis.

- Quantitative methods would benefit from: more details on the variable definitions including poor treatment outcomes (you may consider including a boxed glossary); highlighting missing data in main text and tables; the addition of numerator/denominator for each variable and the outputs of univariable regression in Table 3 (see reviewer 2's comments); additional analyses of social risk factor as a binary variable (0 vs 1 or more) or as 0 vs 1 vs 2 or more in order to allow comparison with related publications (see Reviewer 2's comments); and explanation on the rationale behind why variables such as smoking, smear grade, time from symptom onset to diagnosis, and comorbidities were not included as covariates in the models despite being known to be associated with treatment outcomes.

Thank you for submitting your manuscript to PLOS One and we look forward to receiving the resubmission for re-review.

Reviewers' comments:

Reviewer's Responses to Questions

**Comments to the Author**

1. Is the manuscript technically sound, and do the data support the conclusions?

Reviewer #1: No

Reviewer #2: Yes

2. Has the statistical analysis been performed appropriately and rigorously? 

Reviewer #1: I Don't Know

Reviewer #2: Yes

3. Have the authors made all data underlying the findings in their manuscript fully available?

Reviewer #1: No

Reviewer #2: Yes

4. Is the manuscript presented in an intelligible fashion and written in standard English?

Reviewer #1: Yes

Reviewer #2: Yes

5. Review Comments to the Author

Reviewer #1: PLOS One: Vulnerability and tuberculosis treatment outcomes in urban settings in England: a mixed-methods study.

1. The quantitative part of this study is replicating research originally conducted on a London cohort in different settings. Further qualitative research has been conducted but through purposive sampling of patients experiencing diagnostic delay in addition to experiencing poor TB treatment outcomes. However, diagnostic delay was not commented on in the quantitative analysis and these to read as different pieces of work.

2. I am not aware that this has been published elsewhere.

3. INTRODUCTION –

P11 line 99-100 – what are the specific risk factors linked to marginalisation, would help to list these here.

P11 line 103 – You should state or give examples of the heterogenous subgroups you are referring to.

P11 line 113-114 suggests that this work is going to be a local needs assessment – but this then doesn’t seem to be clear throughout the rest of the article so seems a bit confused.

P12 line 118-119 what are the diverse sources of vulnerabilities you are referring to here. You suggest these are different to those of the individuals social risk factors.

P12 p123-124 Your introduction suggests that you will determine healthcare needs and priorities at a local level, but the rest of the article doesn’t deliver this.

P12 Line 126-127 This seems a bit odd as you have discussed the requirement of locally relevant needs assessments but then are intending to combine findings from two different cities without any further reflection on whether these combined findings will remain locally relevant.

I also think it needs to be clear why you are defining healthcare priorities using qualitative methodology now. Would it not make sense to determine who is likely to have a poor treatment outcome first (from your quantitative work) – then purposively sample those individuals you have identified as being at higher risk of poor treatment outcomes.

METHODS –

As mentioned above I am not sure what the justification is for performing this work concurrently and why this has been done and what this adds.

P13 line 151-153 – This states that individuals were selected purposively if they had a diagnostic delay of more than two months or poor TB treatment outcome. There isn’t any earlier justification for this in the introduction. It is unclear why diagnostic delay is being used as part of the eligibility criteria and should be clarified. Diagnostic delay is introduced for the first time here so you need explain why this is being used.

P13 line 155-156 It isn’t clear why you chose to exclude participants with no final treatment outcome - can you state….because x, y and z. As patients that don’t attend follow up appointments are often found so be vulnerable. Wilson R, Winnard Y. Causes, impacts and possible mitigation of non-attendance of appointments within the National Health Service: a literature review. J Health Organ Manag. 2022 Aug 4;ahead-of-print(ahead-of-print). doi: 10.1108/JHOM-11-2021-0425. PMID: 35918282.

P13 line 161-162 – what informed the topic guides and what did they cover? You should follow Consolidated criteria for reporting qualitative research (COREQ) reporting guidelines.

P14 – line 163-164 – it appears that health care professional interviews were conducted as part of a separate service evaluation. It is therefore not clear from what is stated that these would be focused on the research question being explored here – health care needs/priorities. From what has been described they would have focused on evaluating the service being delivered so this requires clarification and data removing if addressing a different issue.

DATA ANAYLSIS –

P14 line 172-173 Some introduction of the poor treatment outcomes you are referring to in this work in the introduction would be helpful to justify their selection. Again, you don’t make any reference to delayed diagnosis and its role however you have used it as a criteria for selection of you interview participants which make this feel like a separate piece of work to that of the quantitative data.

P15 line 191 – You state you used some deductive theory driven analysis – but you don’t state what existing theories you have used to base this analysis on or how this was conducted.

RESULTS –

Table 1 – this details that 40% of the total patients had extrapulmonary TB – at no point has there been any discussion about diagnosis of pulmonary versus extrapulmonary and clinical complexity. This is vital if you are going to discuss and select patients based on diagnostic delay. I appreciate this doesn’t appear to be what this article set out to look at and that is why is seems somewhat confused.

P16 line 216-217 nine patients with TB were interviewed but you don’t state whether this was extrapulmonary or pulmonary TB. If you are going to start discussing diagnostic delay you must do this to put this in context and recognise the clinical complexity of coming to a primary TB diagnosis in a low incidence setting.

This is not a narrative that weaves together. The quantitative and qualitative findings are described independently and they don’t overlap in the themes. As stated previously a consecutive research strategy where you quantitatively identify what your risks are for poor TB treatment outcomes in your population first and then select your participants for qualitative inquiry based on those factors would have been more robust. This currently reads like two independent pieces of work.

P233 social aspects of TB reported here have been previously reported and recognised.

P19 – line 290-292 is it realistic/ sensible to expect TB (in a low incidence setting) to be an immediate diagnosis that a GP considers when a patients attends with symptoms (the majority of patients attend GP surgeries with symptoms). You should at a minimum state – symptoms consistent with a diagnosis of TB. Furthermore, recognise that these can be non-specific at initial presentation and that there can be clinical complexity. It is also not clear how this relates to your primary line of questioning for this paper. The sample here is biased as you stated in your methods that you were selecting patients based on having had a diagnostic delay and I think you need to justify why this was done.

The qualitative results that have been reported do appear to be more in keeping and focused on a service evaluation rather than a needs assessment determining healthcare priories in those with poor TB outcomes.

4. DISCUSSION –

Birmingham and Leicester are two different settings however findings have been amalgamated and no consideration has been given to their differences in terms of interpretation of results. You have in your introduction highlighted the importance of local relevant data so this seems at odds with what you have done.

P23 line 383-367 – vulnerability to developing TB rather than TB treatment outcomes has been introduced as a concept here – which is not what this study initially reported it was setting out to look at. I don’t think you can make these claims based on this body of work as it does not provide sufficient evidence. This is an overstated conclusion and should at a minimum be framed as a hypothesis.

P24 line 407-409 “South Asians form the highest proportion but do not

experience poorer outcomes probably due to their adaption, over several decades, to the healthcare system compared with recently arrived Central Europeans”. this statement requires removal – what evidence are you basing this statement on? This statement has an undertone of discrimination.

P26 limitations – You should comment on combining these data from two different settings – at present there is no reflection on whether that makes sense or is justified. No reflection on the biases of those conducting interviews or the selection process or drop out those approached who declined to be interviewed and why.

The reported qualitative findings don’t meet Consolidated criteria for reporting qualitative research (COREQ) reporting guidelines. https://academic.oup.com/intqhc/article/19/6/349/1791966

Reviewer #2: Thank you for the opportunity to review this manuscript, which used mixed-methods to investigate risk factors for adverse treatment outcomes among people with TB in Birmingham and Leicester in England.

I really enjoyed reading this study and congratulate the authors on making use of data collected as part of routine TB registers and the ETS system to analyse risk factors for adverse treatment outcomes in a large cohort. It was also refreshing and useful to read the qualitative findings of the report, which help to add crucial insight and context. The recognition of the importance of using local data to design specific interventions is extremely welcome and relevant, as is the attention given to addressing the social and economic determinants of TB through a multi-sector response including better integration of health, social care, and social protection services. I have minimal experience of qualitative methodology and analysis and so my review therefore focusses more on the quantitative aspects.

General points

- Suggest review language throughout to use non-stigmatising language where possible (https://www.stoptb.org/words-matter-language-guide), and avoid terms such as alcohol abuse.

Specific points

Introduction

- It would be helpful if the authors could include some data on the local epidemiology of TB in Birmingham and Leicester.

Methods/results

- It would be helpful to include more detail in the main text on what the qualitative interviews explored and how/why the questions were developed as they were. The topic guides that have been included helpfully in the appendix should be signposted to so that people know they are there (I didn’t until I checked the supplement). However, I would be tempted to put all the additional information currently in the supplement (except for table S4) in the main text (table S3 can be usefully combined with table 1; and the additional text in the supplement is brief and could be shortened a bit).

- It would be interesting to see the breakdown of adverse outcomes (i.e. % lost to follow-up versus % stopping treatment versus % death). Did the authors consider investigating risk factors for these outcomes separately? Similarly, some more information on how these outcomes are defined and what they mean would be helpful (e.g. does death refer to death during treatment from any cause or specifically from TB?).

- It looks like the analysis was restricted to adults aged 16 and over? A sentence in the methods stating this would be helpful.

- Although table 1 presents the demographic characteristics, it would be helpful in the text to highlight some of the more striking ones (e.g. 59% male, 62% of patients living in the most deprived 20% of areas of the country)

- For each of the variables in table 1, it would be helpful to include the % of data that were missing. It wasn’t clear if the percentages were calculated using the total as the denominator or the total with data available?

- It would be helpful if more information on variable definitions was included in the main text, e.g. the fact that deprivation is based on the IMD. How were drug and alcohol use defined (current/past etc)? Did imprisonment refer to active incarceration or ever been incarcerated? Readers will want to understand more clearly how these data were collected and what they represent to be able to make judgements about their validity, and many will not be familiar with the UK systems used to collect data on TB notifications.

- The authors have presented an overall multivariable regression model. I think it would be useful also see the results of a univariable unadjusted regression analysis. I also wonder if the tables could be re-organised so that readers can study the absolute proportions with adverse outcomes alongside the ORs for each risk factor (potentially this could be in a new table alongside the unadjusted analysis).

- I wonder if the IMD should be analysed differently. Deprivation is such a strong risk factor for TB and adverse outcomes in other settings (and indeed the ORs presented are suggestive here of an association, especially in Leicester) and I wonder if the small number of people in the least deprived deciles is masking a true effect. I would be interested to see an analysis of a dichotomised variable (e.g. IMD 1-5 versus 6-10).

- Similarly, I wondered if the authors considered an analysis of any social risk factor versus none; or 2 social risk factors versus one versus none (as presented in the UKHSA TB in England report).

- The study took place over many years and care provided to patients may have changed during that time. Was year considered in the analysis as a potential predictor of bad outcome?

- Have the authors considered using population attributable fractions as a way of illustrating the public health importance of these risk factors? This might complement the current analysis nicely.

Discussion

- I think more needs to be included on the limitations of this study, particularly in terms of the quantitative data. For example, the data on social risk factors are potentially subject to significant social desirability bias and thus underreporting. I also personally didn’t completely agree with the way these results were framed as showing no association between social risk factors and adverse outcomes. The numbers of people reporting these risk factors were small, and, because of this, the data presented don’t really provide evidence showing no association between these variables and adverse treatment outcomes as the confidence intervals are so wide. Indeed, for alcohol for example, the OR point estimate is 1.8 and the confidence interval 0.6-4.9. This relates to the point above about PAFs; something can be an important individual risk factor but have much less importance at a population level because of the low population prevalence of the risk factor. It seems more likely and plausible to me that there is an association between alcohol misuse and adverse outcome that would have been detected if the study was larger, but it’s PAF is likely to be very low. I’d be interested to see more discussion around this issue in the manuscript.

- I think it’s also worth mentioning that some of the most vulnerable people (who may well have some of these social risk factors) are the people who are most likely to not have had their TB diagnosed and thus been included in this study.

- Furthermore, I think it would be worth highlighting some other variables that have been shown to be important in other settings for predicting adverse outcomes, for which data were not available/not analysed in this study (e.g. were data on smoking, smear grade, symptom duration before diagnosis, co-morbidities available?).

- I think some mention of the definition of adverse outcome not encapsulating all adverse impacts of TB would be worthwhile (e.g. some of those who completed treatment will have TB recurrence, others will have significant post-TB lung disease…and it’s plausible that social risk factors are particularly important for some of those impacts).

- The point about South Asian people having better adaptation to the UK health system is interesting. Do the authors have any data on length of time in the UK available for their participants? Is there any evidence to support that hypothesis?

- Some comparison of overall treatment success rates to the UK average would be interesting.

- Overall the discussion is very good and brings together the different aspects of the manuscript nicely to provide recommendations for action.

6. PLOS authors have the option to publish the peer review history of their article (what does this mean?). If published, this will include your full peer review and any attached files.

Reviewer #1: No

Reviewer #2: **Yes: **Matthew Saunders

---

## [Author Response · Author response to Decision Letter 0]

24 Jan 2023

Reviewer #1: PLOS One: Vulnerability and tuberculosis treatment outcomes in urban settings in England: a mixed-methods study.

Point 1. The quantitative part of this study is replicating research originally conducted on a London cohort in different settings. Further qualitative research has been conducted but through purposive sampling of patients experiencing diagnostic delay in addition to experiencing poor TB treatment outcomes. However, diagnostic delay was not commented on in the quantitative analysis and these to read as different pieces of work.

Reply: The reviewer is right that this study replicates some work done in London, however, few studies have been performed outside of London addressing this topic, as a result, the epidemiology of TB in London has shaped the national TB control agenda in England. On the other hand, the main focus of this study was not to understand the phenomenon of diagnostic delay but to get insights into healthcare priorities, service improvement and novel interventions that might be most appropriate for the target group taking into account its setting specific characteristics. Diagnostic delay was used as a variable to select participants for the interviews because according to the epidemiology in England underserved populations are mainly affected by diagnostic delay. The reviewer is right that there was not enough mention or clarification on diagnostic delay in the first version of the manuscript. In the revised version we introduced diagnostic delay early on in the introduction in line 104

Point 2. I am not aware that this has been published elsewhere.

Reply: The study has not been published elsewhere

3. INTRODUCTION –

Point 3. P11 line 99-100 – what are the specific risk factors linked to marginalisation, would help to list these here.

Reply: Two examples were added 

Point 4. P11 line 103 – You should state or give examples of the heterogenous subgroups you are referring to.

Reply: The heterogenous subgroups are mentioned in the following sentence of that paragraph lines 104-106 “In England, underserved populations are identified by the presence of predefined social risk factors (SRFs) (i.e. drug or alcohol abuse, homelessness, imprisonment)”

Point 5. P11 line 113-114 suggests that this work is going to be a local needs assessment – but this then doesn’t seem to be clear throughout the rest of the article so seems a bit confused.

Reply: Thanks to the reviewer for rising this important and constructive point, the manuscript was amended to highlight this aspect in lines 248-252 and 408-418. This can also be appreciated with the topic guides which were provided in the first submission but were not easy to find. In the revised version the topic guides were brought to the main text Tables 1 and 2.

Point 6: P12 line 118-119 what are the diverse sources of vulnerabilities you are referring to here. You suggest these are different to those of the individuals social risk factors.

Reply: Examples were added in line 117 “interrelated vulnerabilities e.g., income distribution or lack of social capital”.

Point 7: P12 p123-124 Your introduction suggests that you will determine healthcare needs and priorities at a local level, but the rest of the article doesn’t deliver this.

Reply: We think this point is related to reviewer’s previous point 5. We added additional text highlighting this aspect in the introduction section lines 129-130, how it was addressed in the qualitative part of the study in results section lines 249-253 and a summary of key findings in discussion section lines 416-419. The reviewer can also check the topic guides which are now in the main text as Tables 1 and 2.

Point 8: P12 Line 126-127 This seems a bit odd as you have discussed the requirement of locally relevant needs assessments but then are intending to combine findings from two different cities without any further reflection on whether these combined findings will remain locally relevant.

I also think it needs to be clear why you are defining healthcare priorities using qualitative methodology now. Would it not make sense to determine who is likely to have a poor treatment outcome first (from your quantitative work) – then purposively sample those individuals you have identified as being at higher risk of poor treatment outcomes.

Reply: We do not agree with the reviewer that the results are combined. Tables 1-3 present results for the two cities independently and then in a column representing the full study cohort from the two cities. Thus, the reader can view both sets of results. Likewise, in the text, the results for the two cities are presented and we highlight that they were similar, lines 214-217. We did not find any difference that merit an independent presentation or discussion. The same applies to the qualitative results, the results were similar, had we found any difference between the two cities we would have presented it in the results, but it was not the case.

Regarding the use of qualitative methods to define healthcare priorities. This is simply something that cannot be done using only quantitative methods. In lines 121-123 of the introduction, we mention that “Most studies on TB treatment outcomes in underserved populations in England have been done in London, however this may not be representative of the national picture” For this reason, in terms of TB control, the epidemiology of TB in London has driven control policy in England based mainly on a few quantitative studies, and this is exactly what this study aims to address. We also mentioned in lines 114-116 of the introduction that “intervention appropriateness should be informed by a local needs assessment. For this, the target population must be first defined and characterised to enable the identification of their healthcare priorities.”

On the selection of participants for the qualitative study, we think that both approaches are valid, the one the reviewer proposed and the one we used. We selected patients based on variables known to be related to poor treatment outcome and diagnostic delay. The process was guided by the local lead TB nurses who were very familiar with their patient populations. In fact, had we followed the approach suggested by the reviewer we would most probably have ended up interviewing the same patients because we were constrained by patients willingness to be interviewed, this was a major challenge. According to reviewer’s comment we added a mention of this limitation in the discussion section lines 508-510

METHODS –

Point 9: As mentioned above I am not sure what the justification is for performing this work concurrently and why this has been done and what this adds.

Reply: Our answer to point 8 address this point.

Point 10: P13 line 151-153 – This states that individuals were selected purposively if they had a diagnostic delay of more than two months or poor TB treatment outcome. There isn’t any earlier justification for this in the introduction. It is unclear why diagnostic delay is being used as part of the eligibility criteria and should be clarified. Diagnostic delay is introduced for the first time here so you need explain why this is being used.

Reply: Diagnostic delay was used because according to published literature, most underserved populations in England suffer diagnostic delays and this is one of the key variables used by the UK Health Security Agency to identify underserved populations. For this reason, one of the key goals of the collaborative strategy for TB control in England 2015-2020, was to reduce diagnostic delay in underserved populations. We provide references 1 and 4 which contain further information about the topic. The reviewer is right that it should have been highlighted earlier in the introduction, we amended the manuscript and mention it in line 104.

Point 11: P13 line 155-156 It isn’t clear why you chose to exclude participants with no final treatment outcome - can you state….because x, y and z. As patients that don’t attend follow up appointments are often found so be vulnerable. Wilson R, Winnard Y. Causes, impacts and possible mitigation of non-attendance of appointments within the National Health Service: a literature review. J Health Organ Manag. 2022 Aug 4;ahead-of-print(ahead-of-print). doi: 10.1108/JHOM-11-2021-0425. PMID: 35918282.

Reply: The patients that did not have available information on final treatment outcome were excluded because it was the primary outcome of the quantitative analysis. Therefore we could not include them in the analysis. We mention it in the manuscript in line 156 as requested by the reviewer.

Point 12: P13 line 161-162 – what informed the topic guides and what did they cover? You should follow Consolidated criteria for reporting qualitative research (COREQ) reporting guidelines.

Reply: The topic guides had been included in the supplement of the first submission. They are now in the main text Tables 1 and 2. The topic guides were based on discussions with healthcare professionals and informed by the goals of the Collaborative Tuberculosis Strategy for England 2015-20 as mentioned in lines 171-173 of the revised manuscript.

Point 13: P14 – line 163-164 – it appears that health care professional interviews were conducted as part of a separate service evaluation. It is therefore not clear from what is stated that these would be focused on the research question being explored here – health care needs/priorities. From what has been described they would have focused on evaluating the service being delivered so this requires clarification and data removing if addressing a different issue.

Reply: The interviews with healthcare professionals were not conducted as a separate work. The topic guides had been included in the first submission. They’re more visible in the revised manuscript and the reviewer can refer to them to see that they addressed the same topic. Under the remit of the UK Health Security Agency we did not need ethics approval to talk to healthcare professionals, for this reason we started the interviews with them while we waited for ethics approval to talk with patients. 

DATA ANAYLSIS –

Point 14: P14 line 172-173 Some introduction of the poor treatment outcomes you are referring to in this work in the introduction would be helpful to justify their selection. Again, you don’t make any reference to delayed diagnosis and its role however you have used it as a criteria for selection of you interview participants which make this feel like a separate piece of work to that of the quantitative data.

Reply: The treatment outcomes are the same ones used by the UK Health Security Agency, we mentioned this in the revised manuscript and cited reference 1. Regarding diagnostic delay, we addressed this in our reply to reviewer’s point 10.

Point 15: P15 line 191 – You state you used some deductive theory driven analysis – but you don’t state what existing theories you have used to base this analysis on or how this was conducted.

Reply: Thanks again to the reviewer for highlighting this very important point that we forgot to mention in the first version of the manuscript. The revised manuscript was amended with new information on this aspect in lines 203-207.

RESULTS –

Point 16: Table 1 – this details that 40% of the total patients had extrapulmonary TB – at no point has there been any discussion about diagnosis of pulmonary versus extrapulmonary and clinical complexity. This is vital if you are going to discuss and select patients based on diagnostic delay. I appreciate this doesn’t appear to be what this article set out to look at and that is why is seems somewhat confused.

Reply: We addressed the points related to diagnostic delay previously in points 1 and 10. We think that a discussion on the diagnosis of pulmonary versus extrapulmonary TB is out of the scope of this study.

Point 17: P16 line 216-217 nine patients with TB were interviewed but you don’t state whether this was extrapulmonary or pulmonary TB. If you are going to start discussing diagnostic delay you must do this to put this in context and recognise the clinical complexity of coming to a primary TB diagnosis in a low incidence setting.

Reply: This is another important omission from our part, in the revised version we mention that all patients were diagnosed with pulmonary TB in line 231-232. We also mention as a limitation that no patients with extrapulmonary TB were interviewed in lines 493-495 of the discussion.

Point 18: This is not a narrative that weaves together. The quantitative and qualitative findings are described independently and they don’t overlap in the themes. As stated previously a consecutive research strategy where you quantitatively identify what your risks are for poor TB treatment outcomes in your population first and then select your participants for qualitative inquiry based on those factors would have been more robust. This currently reads like two independent pieces of work.

Reply: This point has been addressed in our replies to reviewer’s points 5,7-10,12 and 13

Point 19: P233 social aspects of TB reported here have been previously reported and recognised.

Reply: The reviewer must be referring to some work done in London, however, few to no information exist outside London. 

Point 20: P19 – line 290-292 is it realistic/ sensible to expect TB (in a low incidence setting) to be an immediate diagnosis that a GP considers when a patients attends with symptoms (the majority of patients attend GP surgeries with symptoms). You should at a minimum state – symptoms consistent with a diagnosis of TB. Furthermore, recognise that these can be non-specific at initial presentation and that there can be clinical complexity. It is also not clear how this relates to your primary line of questioning for this paper. The sample here is biased as you stated in your methods that you were selecting patients based on having had a diagnostic delay and I think you need to justify why this was done.

Reply: The study was done in high incidence areas in Birmingham and Leicester, like in some boroughs of London, the incidence of TB is very high in some areas of these two cities. For this reason, the goals of the collaborative Collaborative Tuberculosis Strategy for England 2015–2020 included to improve access to services and ensure early diagnosis and to ensure an appropriate workforce to deliver TB control, further information can be accessed in reference 5. Within this context, the NHS and the UKHSA started a programme to train GPs in these areas to recognise and diagnose TB. Thus, we think it is pertinent and relevant that TB patients suggest that GPs in these areas should be able to recognise and diagnose TB. The results of this study have already informed the strengthening of training for GP in high incidence areas of Birmingham and Leicester led by the NHS and the UKHSA. 

We addressed the point related to diagnostic delay in our reply to points 1 and 10.

Point 21: The qualitative results that have been reported do appear to be more in keeping and focused on a service evaluation rather than a needs assessment determining healthcare priories in those with poor TB outcomes.

Reply: We addressed this point in our reply to points 5, 7-9, 13 and 14

4.DISCUSSION –

Point 22: Birmingham and Leicester are two different settings however findings have been amalgamated and no consideration has been given to their differences in terms of interpretation of results. You have in your introduction highlighted the importance of local relevant data so this seems at odds with what you have done.

Reply: Although Birmingham and Leicester are different settings, they share similar characteristics in terms of TB epidemiology and demographic make-up with large communities with heritage from the Indian subcontinent and TB concentrated in non-UK born individuals. We included a brief mention about this in the revised version line 127 of the introduction and cited reference 1. Likewise, we highlight in the results lines 261-263 that the qualitative results in both settings were similar and for that reason they’re not presented separately. 

Point 23: P23 line 383-367 – vulnerability to developing TB rather than TB treatment outcomes has been introduced as a concept here – which is not what this study initially reported it was setting out to look at. I don’t think you can make these claims based on this body of work as it does not provide sufficient evidence. This is an overstated conclusion and should at a minimum be framed as a hypothesis.

Reply: The word vulnerability was deleted in line 430.

Point 24: P24 line 407-409 “South Asians form the highest proportion but do not

experience poorer outcomes probably due to their adaption, over several decades, to the healthcare system compared with recently arrived Central Europeans”. this statement requires removal – what evidence are you basing this statement on? This statement has an undertone of discrimination.

Reply: Five of our co-authors are healthcare professionals with heritage from the Indian sub-continent who were born, raised, educated and work in Birmingham or Leicester. They proposed this idea in our discussion of the results. We rephrased it as a hypothesis, and although we do not agree with the reviewer that this may have an undertone of discrimination, we changed the sentence so that we do not name any particular group.

Point 25: P26 limitations – You should comment on combining these data from two different settings – at present there is no reflection on whether that makes sense or is justified. No reflection on the biases of those conducting interviews or the selection process or drop out those approached who declined to be interviewed and why.

Reply: We addressed the first part of this point on the form of presenting the results in our answer to point 22. The revised manuscript mention the biases of interviewers and the selection process in lines 169-171 of the methods section and lines 501-503 of the discussion

Point 26: The reported qualitative findings don’t meet Consolidated criteria for reporting qualitative research (COREQ) reporting guidelines. https://academic.oup.com/intqhc/article/19/6/349/1791966

Reply: The items that we forgot to mention and that the reviewer spotted were introduced in the revised version, mainly the relationship with participants and theoretical framework

Reviewer #2: Thank you for the opportunity to review this manuscript, which used mixed-methods to investigate risk factors for adverse treatment outcomes among people with TB in Birmingham and Leicester in England.

I really enjoyed reading this study and congratulate the authors on making use of data collected as part of routine TB registers and the ETS system to analyse risk factors for adverse treatment outcomes in a large cohort. It was also refreshing and useful to read the qualitative findings of the report, which help to add crucial insight and context. The recognition of the importance of using local data to design specific interventions is extremely welcome and relevant, as is the attention given to addressing the social and economic determinants of TB through a multi-sector response including better integration of health, social care, and social protection services. I have minimal experience of qualitative methodology and analysis and so my review therefore focusses more on the quantitative aspects.

Point 1: General points

- Suggest review language throughout to use non-stigmatising language where possible (https://www.stoptb.org/words-matter-language-guide), and avoid terms such as alcohol abuse.

Reply: We changed TB control for TB prevention and care, and drug or alcohol abuse for drug or alcohol problem throughout the manuscript.

Specific points

Introduction

Point 2: - It would be helpful if the authors could include some data on the local epidemiology of TB in Birmingham and Leicester.

Reply: We mention in lines 127-129 that the two cities have similar epidemiological patterns, the TB rates and the main groups affected by TB

Methods/results

Point 3- It would be helpful to include more detail in the main text on what the qualitative interviews explored and how/why the questions were developed as they were. The topic guides that have been included helpfully in the appendix should be signposted to so that people know they are there (I didn’t until I checked the supplement). However, I would be tempted to put all the additional information currently in the supplement (except for table S4) in the main text (table S3 can be usefully combined with table 1; and the additional text in the supplement is brief and could be shortened a bit).

Reply: We included more information on how the topic guides were developed in lines 173-175 and moved the topic guides to the main text, tables 1 and 2. This will make it easy for the reader to put the qualitative results in context according to the questions. This is in line with some points from reviewer 1. Regarding table S1, we would recommend to leave it in the supplement so that it can be read together with the text provided on multiple imputation. It is a technique difficult to implement, so we want to provide the readers with all details on how the models were built and what type of regression was used for each variable.

Point 4: - It would be interesting to see the breakdown of adverse outcomes (i.e. % lost to follow-up versus % stopping treatment versus % death). Did the authors consider investigating risk factors for these outcomes separately? Similarly, some more information on how these outcomes are defined and what they mean would be helpful (e.g. does death refer to death during treatment from any cause or specifically from TB?).

Reply: The percentage of each outcome was added to table 3 as suggested by the reviewer. Death included; TB as cause of death, TB contributed to death, TB incidental to death and relationship between TB and death unknown. This was added to the manuscript in lines 190-191

Point 5:- It looks like the analysis was restricted to adults aged 16 and over? A sentence in the methods stating this would be helpful.

Reply: This is now mentioned in the methods section line 144 

Point 6:- Although table 1 presents the demographic characteristics, it would be helpful in the text to highlight some of the more striking ones (e.g. 59% male, 62% of patients living in the most deprived 20% of areas of the country)

Reply: This additional percentage was added to the main text as suggested lines 231-232

Point 7:- For each of the variables in table 1, it would be helpful to include the % of data that were missing. It wasn’t clear if the percentages were calculated using the total as the denominator or the total with data available?

Reply: In the revised version we highlighted that details of percentage of missing information is available in S1 Table of the supporting information lines 196-197. The percentage is presented for the total missing information for each variable.

Point 8:- It would be helpful if more information on variable definitions was included in the main text, e.g. the fact that deprivation is based on the IMD. How were drug and alcohol use defined (current/past etc)? Did imprisonment refer to active incarceration or ever been incarcerated? Readers will want to understand more clearly how these data were collected and what they represent to be able to make judgements about their validity, and many will not be familiar with the UK systems used to collect data on TB notifications.

Reply: In the revised version the supporting information is signposted so that it is easier to find, it is also highlighted at the end of the text. The supporting information contains details on the IMD. In addition, we added to the supporting information that the social risk factors were categorised as ever experienced one of those.

Point 9:- The authors have presented an overall multivariable regression model. I think it would be useful also see the results of a univariable unadjusted regression analysis. I also wonder if the tables could be re-organised so that readers can study the absolute proportions with adverse outcomes alongside the ORs for each risk factor (potentially this could be in a new table alongside the unadjusted analysis).

Reply: We added the univariate analysis to the supplementary information Table S2 and mentioned in the main text that the results of the univariate analysis are available in the supplement line 264. However, we did not add the absolute proportions for each variable because this information is already available in Table 1.

Point 10:- I wonder if the IMD should be analysed differently. Deprivation is such a strong risk factor for TB and adverse outcomes in other settings (and indeed the ORs presented are suggestive here of an association, especially in Leicester) and I wonder if the small number of people in the least deprived deciles is masking a true effect. I would be interested to see an analysis of a dichotomised variable (e.g. IMD 1-5 versus 6-10).

Reply: We had reported previously a significant association between deprivation and the risk of developing TB using population-based data of all England Berrocal-Almanza L et al. 2019. PMID:31471131 DOI:10.1016/S1473-3099(19)30260-9. We’re not aware of evidence on the association of IMD and treatment outcomes in England. However, would advise against aggregating the data in only two categories because it would not take into account all nuances around the intermediate categories 4-6 that do not belong neither to the most nor the least deprived. We followed reviewer’s recommendation and ran a new analysis using the same categories used in our 2019 article; 1-3 most deprived, 4-6 and 7-10 least deprived, but we did not see any significant association.

Point 11:- Similarly, I wondered if the authors considered an analysis of any social risk factor versus none; or 2 social risk factors versus one versus none (as presented in the UKHSA TB in England report).

Reply: Yes, we thought about it but the frequency of individuals with two or more risk factors in our cohort was extremely low and for this reason no meaningful analysis could be done.

Point 12: - The study took place over many years and care provided to patients may have changed during that time. Was year considered in the analysis as a potential predictor of bad outcome?

Reply: Yes, we considered year, it was introduced in an univariate model and we did not find any significant association. It was not included in the multivariate model because the inclusion of this variable did not improve the goodness of fit of the overall model.

Point 13:- Have the authors considered using population attributable fractions as a way of illustrating the public health importance of these risk factors? This might complement the current analysis nicely.

Reply: We thought about it, however, the prevalence of these factors in Birmingham and Leicester is very low and according to the healthcare providers and patients we interviewed they do not determine much the treatment outcomes. We think that the factors may be very important to influence outcomes in settings such as London but they are not so relevant in places like Birmingham or Leicester and our results both quantitative and qualitative support that conclusion.

Discussion

Point 14: - I think more needs to be included on the limitations of this study, particularly in terms of the quantitative data. For example, the data on social risk factors are potentially subject to significant social desirability bias and thus underreporting. I also personally didn’t completely agree with the way these results were framed as showing no association between social risk factors and adverse outcomes. The numbers of people reporting these risk factors were small, and, because of this, the data presented don’t really provide evidence showing no association between these variables and adverse treatment outcomes as the confidence intervals are so wide. Indeed, for alcohol for example, the OR point estimate is 1.8 and the confidence interval 0.6-4.9. This relates to the point above about PAFs; something can be an important individual risk factor but have much less importance at a population level because of the low population prevalence of the risk factor. It seems more likely and plausible to me that there is an association between alcohol misuse and adverse outcome that would have been detected if the study was larger, but it’s PAF is likely to be very low. I’d be interested to see more discussion around this issue in the manuscript.

Reply: As per reviewer’s comment, we tone downed our mention on no association by deleting this from our conclusions lines 66-67 of the abstract and lines 426-427 of the discussion. However, we ask the reviewer to take into account that, although as the reviewer mentions the addition of more data would potentially show an association, such data for these two settings does not exist because we have the top prevalence for the study period. There’s simply no more patients to add to the cohort for Birmingham and Leicester, and TB is a disease of mandatory notification. We could of course add more data from other parts of England, let’s say we add London data, such an association may appear but the London data would be skewing the distribution because the association would only exist in London, not in Birmingham or Leicester. That difference would appear if a stratified analysis by location is done. Such stratified analysis would reveal that this association only holds for London. That’s exactly the reason we did this study, because the UKHSA has been analysing aggregated data for decades without taking into account regional variation. As a result, the epidemiology of TB in London has driven TB prevention and care policy. We had already mentioned that we could not rule out an association in lines 450-452. However, based on the explanation we provided above, we think that statement should be removed.

Point 15: - I think it’s also worth mentioning that some of the most vulnerable people (who may well have some of these social risk factors) are the people who are most likely to not have had their TB diagnosed and thus been included in this study.

Reply: We included a mention on this in lines 505-507 of the discussion section.

Point 16:- Furthermore, I think it would be worth highlighting some other variables that have been shown to be important in other settings for predicting adverse outcomes, for which data were not available/not analysed in this study (e.g. were data on smoking, smear grade, symptom duration before diagnosis, co-morbidities available?).

Reply: From the variables that the reviewer mentions only smoking and co-morbidities were not available. Smear test was included in the univariate analysis now shown in Table S2, however, it did not contribute to improve the fit of the multivariate model and therefore we dropped it. We mention the no inclusion of co-morbidities in lines 500-502 of the revised manuscript.

Point 17:- I think some mention of the definition of adverse outcome not encapsulating all adverse impacts of TB would be worthwhile (e.g. some of those who completed treatment will have TB recurrence, others will have significant post-TB lung disease…and it’s plausible that social risk factors are particularly important for some of those impacts).

Reply: Although this is something relevant, we respectfully think it is out of the scope of this study.

Point 18:- The point about South Asian people having better adaptation to the UK health system is interesting. Do the authors have any data on length of time in the UK available for their participants? Is there any evidence to support that hypothesis?

Reply: Five of our co-authors are healthcare professionals with heritage from the Indian sub-continent who were born, raised, educated and work in Birmingham or Leicester. They proposed this idea in our discussion of the results. We rephrased it as a hypothesis because we do not have any data to support this claim.

Point 19:- Some comparison of overall treatment success rates to the UK average would be interesting.

Reply: Although this is something that could be interesting as the reviewer mentions, our manuscript is already in 4,223 words after all the additions recommended by both reviewers. For this reason, we think this is something we could leave out.

Point 20:- Overall the discussion is very good and brings together the different aspects of the manuscript nicely to provide recommendations for action.

Reply: The manuscript overall improved with the additions recommended by both reviewers, thank you for taking the time to read our work and contribute to improve it.

---

## [Decision Letter · Decision Letter 1]

30 Jan 2023

PONE-D-22-28286R1

Vulnerability and tuberculosis treatment outcomes in urban settings in England: a mixed-methods study.

PLOS ONE

Dear Dr Berrocal-Almanza,

Thank you for submitting your manuscript to PLOS ONE. After careful consideration, we feel that it has merit but does not fully meet PLOS ONE’s publication criteria as it currently stands. Therefore, we invite you to submit a revised version of the manuscript that addresses the points raised during the review process.

Thank you for your revision. I would appreciate it if you were able to respond to Reviewer 1's remaining minor comments. I am confident that, following these minor revisions, we will then be able to accept your manuscript.

If applicable, we recommend that you deposit your laboratory protocols in protocols.io to enhance the reproducibility of your results. Protocols.io assigns your protocol its own identifier (DOI) so that it can be cited independently in the future. For instructions see: https://journals.plos.org/plosone/s/submission-guidelines#loc-laboratory-protocols. Additionally, PLOS ONE offers an option for publishing peer-reviewed Lab Protocol articles, which describe protocols hosted on protocols.io. Read more information on sharing protocols at https://plos.org/protocols?utm_medium=editorial-emailutm_source=authorlettersutm_campaign=protocols.

We look forward to receiving your revised manuscript.

Kind regards,

Tom E. Wingfield

Academic Editor

PLOS ONE

Journal Requirements:

Reviewers' comments:

Reviewer's Responses to Questions

**Comments to the Author**

1. If the authors have adequately addressed your comments raised in a previous round of review and you feel that this manuscript is now acceptable for publication, you may indicate that here to bypass the “Comments to the Author” section, enter your conflict of interest statement in the “Confidential to Editor” section, and submit your "Accept" recommendation.

Reviewer #1: (No Response)

Reviewer #2: All comments have been addressed

2. Is the manuscript technically sound, and do the data support the conclusions?

Reviewer #1: Partly

Reviewer #2: Yes

3. Has the statistical analysis been performed appropriately and rigorously? 

Reviewer #1: Yes

Reviewer #2: Yes

4. Have the authors made all data underlying the findings in their manuscript fully available?

Reviewer #1: Yes

Reviewer #2: Yes

5. Is the manuscript presented in an intelligible fashion and written in standard English?

Reviewer #1: Yes

Reviewer #2: Yes

6. Review Comments to the Author

Reviewer #1: Thank you for responding to many of the points raised regarding your article. I still feel some points have not been fully addressed.

Point 1. The quantitative part of this study is replicating research originally conducted

on a London cohort in different settings. Further qualitative research has been

conducted but through purposive sampling of patients experiencing diagnostic delay in

addition to experiencing poor TB treatment outcomes. However, diagnostic delay was

not commented on in the quantitative analysis and these to read as different pieces of

work.

Reply: The reviewer is right that this study replicates some work done in London,

however, few studies have been performed outside of London addressing this topic, as

a result, the epidemiology of TB in London has shaped the national TB control agenda

in England. On the other hand, the main focus of this study was not to understand the

phenomenon of diagnostic delay but to get insights into healthcare priorities, service

improvement and novel interventions that might be most appropriate for the target

group taking into account its setting specific characteristics. Diagnostic delay was used

as a variable to select participants for the interviews because according to the

epidemiology in England underserved populations are mainly affected by diagnostic

delay. The reviewer is right that there was not enough mention or clarification on

diagnostic delay in the first version of the manuscript. In the revised version we

introduced diagnostic delay early on in the introduction in line 104

Further reply: Thank you for clarifying perhaps states in abstract p3 48-49-“urban area of England other than London”

Also I think you should clarify diagnostic delay in the abstract if possible.

Point 8: P12 Line 126-127 This seems a bit odd as you have discussed the

requirement of locally relevant needs assessments but then are intending to combine

findings from two different cities without any further reflection on whether these

combined findings will remain locally relevant.

I also think it needs to be clear why you are defining healthcare priorities using

qualitative methodology now. Would it not make sense to determine who is likely to

have a poor treatment outcome first (from your quantitative work) – then purposively

sample those individuals you have identified as being at higher risk of poor treatment

outcomes.

Reply: We do not agree with the reviewer that the results are combined. Tables 1-3

present results for the two cities independently and then in a column representing the

full study cohort from the two cities. Thus, the reader can view both sets of results.

Likewise, in the text, the results for the two cities are presented and we highlight that

they were similar, lines 214-217. We did not find any difference that merit an

independent presentation or discussion. The same applies to the qualitative results,

the results were similar, had we found any difference between the two cities we would

have presented it in the results, but it was not the case.

Further reply: If this is the case perhaps a sentence to state there were no significant differences in findings between Leceister and Birmingham would be sufficient but just eyeballing there seems to be a much greater percentage of patients in Birmingham in IMD decile 1-2 than Leciester 73.6% versus 37.7%. Also Leicester cohort had no homeless or imprisoned patients included.

Regarding the use of qualitative methods to define healthcare priorities. This is simply

something that cannot be done using only quantitative methods. In lines 121-123 of the

introduction, we mention that “Most studies on TB treatment outcomes in underserved

populations in England have been done in London, however this may not be

representative of the national picture” For this reason, in terms of TB control, the

epidemiology of TB in London has driven control policy in England based mainly on a

few quantitative studies, and this is exactly what this study aims to address. We also

mentioned in lines 114-116 of the introduction that “intervention appropriateness

should be informed by a local needs assessment. For this, the target population must

be first defined and characterised to enable the identification of their healthcare

priorities.”

Further Reply: I think there is a bit of confusion here. I am not suggesting that you define the healthcare priorities using quantitative methods but that purely you could have used it to facilitate purpose sampling.

On the selection of participants for the qualitative study, we think that both approaches

are valid, the one the reviewer proposed and the one we used. We selected patients

based on variables known to be related to poor treatment outcome and diagnostic

delay. The process was guided by the local lead TB nurses who were very familiar with

their patient populations. In fact, had we followed the approach suggested by the

reviewer we would most probably have ended up interviewing the same patients

because we were constrained by patients willingness to be interviewed, this was a

major challenge. According to reviewer’s comment we added a mention of this

limitation in the discussion section lines 508-510

Further reply: Yes agreed patients willingness to be interviewed is certainly challenging and a constraint that has to be worked within. I do think that you need to recognise that there are biases to the sampling strategy led by a TB nurse given their specific role in this context. This is part of the COREQ guidance.

METHODS –

Point 9: As mentioned above I am not sure what the justification is for performing this

work concurrently and why this has been done and what this adds.

Reply: Our answer to point 8 address this point.

Further reply: This has not been justified yet. Just a short statement as to what is the advantage of performing this concurrently would do? such as - Did you gain data in realtime to inform your topic guides and subsequently shape your interviews?

Point 16: Table 1 – this details that 40% of the total patients had extrapulmonary TB –

at no point has there been any discussion about diagnosis of pulmonary versus

extrapulmonary and clinical complexity. This is vital if you are going to discuss and

select patients based on diagnostic delay. I appreciate this doesn’t appear to be what

this article set out to look at and that is why is seems somewhat confused.

Reply: We addressed the points related to diagnostic delay previously in points 1 and

10.

We think that a discussion on the diagnosis of pulmonary versus extrapulmonary

TB is out of the scope of this study.

Further reply: If you are going to touch on diagnostic delay in the context of TB presenting to a primary care setting. You need to put in a sentence that recognises that there is diagnostic complexity here. Especially when you are including extrapulmonary TB in your data set. If you feel it is outside of the scope of this article then I would remove the extra-pulmonary data.

Point 20: P19 – line 290-292 is it realistic/ sensible to expect TB (in a low incidence

setting) to be an immediate diagnosis that a GP considers when a patient attends with

symptoms (the majority of patients attend GP surgeries with symptoms). You should at

a minimum state – symptoms consistent with a diagnosis of TB. Furthermore,

recognise that these can be non-specific at initial presentation and that there can be

clinical complexity. It is also not clear how this relates to your primary line of

questioning for this paper. The sample here is biased as you stated in your methods

that you were selecting patients based on having had a diagnostic delay and I think

you need to justify why this was done.

Reply: The study was done in high incidence areas in Birmingham and Leicester, like

in some boroughs of London, the incidence of TB is very high in some areas of these

two cities. For this reason, the goals of the collaborative Collaborative Tuberculosis

Strategy for England 2015–2020 included to improve access to services and ensure

early diagnosis and to ensure an appropriate workforce to deliver TB control, further

information can be accessed in reference 5. Within this context, the NHS and the

UKHSA started a programme to train GPs in these areas to recognise and diagnose

TB. Thus, we think it is pertinent and relevant that TB patients suggest that GPs in

these areas should be able to recognise and diagnose TB. The results of this study

have already informed the strengthening of training for GP in high incidence areas of

Birmingham and Leicester led by the NHS and the UKHSA.

Further reply: Can I please clarify are you suggesting that a patient presenting with any symptom (not symptoms consistent with TB) –low mood, vaginal bleeding, infected CS scar, traumatic injuries, toothache, haemorrhoids, eczema etc etc – the list can continue. Should immediately be considered for a diagnosis of TB? I do agree with you we need to think about this more and in high incidence settings, but I think stating symptoms consistent with TB would be a much more sensible considered approach. I think you need to change the wording of page 37 line 348-350

Furthermore, are the GP surgeries you are referring to in these data practicing in the areas of high incidence you are referring to? Are you able to back this up with your data?

4.DISCUSSION –

Point 22: Birmingham and Leicester are two different settings however findings have

been amalgamated and no consideration has been given to their differences in terms

of interpretation of results. You have in your introduction highlighted the importance of

local relevant data so this seems at odds with what you have done.

Reply: Although Birmingham and Leicester are different settings, they share similar

characteristics in terms of TB epidemiology and demographic make-up with large

communities with heritage from the Indian subcontinent and TB concentrated in non-

UK born individuals. We included a brief mention about this in the revised version line

127 of the introduction and cited reference 1. Likewise, we highlight in the results lines

261-263 that the qualitative results in both settings were similar and for that reason

they’re not presented separately.

Further reply: As stated above there are differences in percentages of homeless and imprisoned patients between the two cities and these are underserved groups you are highlighting in your introduction as important.

Point 26: The reported qualitative findings don’t meet Consolidated criteria for reporting

qualitative research (COREQ) reporting guidelines.

https://academic.oup.com/intqhc/article/19/6/349/1791966

Reply: The items that we forgot to mention and that the reviewer spotted were

introduced in the revised

Further reply: The reporting doesn’t currently meet all COREQ reporting guidance please see checklist.

https://cdn.elsevier.com/promis_misc/ISSM_COREQ_Checklist.pdf

Reviewer #2: (No Response)

7. PLOS authors have the option to publish the peer review history of their article (what does this mean?). If published, this will include your full peer review and any attached files.

Reviewer #1: No

Reviewer #2: **Yes: **Matthew Saunders

---

## [Author Response · Author response to Decision Letter 1]

2 Feb 2023

Second revision point-by-point response

6. Review Comments to the Author

Reviewer #1: Thank you for responding to many of the points raised regarding your article. I still feel some points have not been fully addressed.

Second revision points:

Point 1. The quantitative part of this study is replicating research originally conducted

on a London cohort in different settings. Further qualitative research has been

conducted but through purposive sampling of patients experiencing diagnostic delay in

addition to experiencing poor TB treatment outcomes. However, diagnostic delay was

not commented on in the quantitative analysis and these to read as different pieces of

work.

Reply: The reviewer is right that this study replicates some work done in London,

however, few studies have been performed outside of London addressing this topic, as

a result, the epidemiology of TB in London has shaped the national TB control agenda

in England. On the other hand, the main focus of this study was not to understand the

phenomenon of diagnostic delay but to get insights into healthcare priorities, service

improvement and novel interventions that might be most appropriate for the target

group taking into account its setting specific characteristics. Diagnostic delay was used

as a variable to select participants for the interviews because according to the

epidemiology in England underserved populations are mainly affected by diagnostic

delay. The reviewer is right that there was not enough mention or clarification on

diagnostic delay in the first version of the manuscript. In the revised version we

introduced diagnostic delay early on in the introduction in line 104

Further reply: Thank you for clarifying perhaps states in abstract p3 48-49-“urban area of England other than London”

Also I think you should clarify diagnostic delay in the abstract if possible.

Reply: We added to the abstract “urban areas of England other than London” in line 49 of the abstract as recommended. Regarding diagnostic delay, we could not find a way to clarify this without making the abstract too long. A midpoint solution we propose is to highlight in the abstract that the interviewed participants were purposely selected. We included this in line 55-46 of the abstract. 

Point 8: P12 Line 126-127 This seems a bit odd as you have discussed the

requirement of locally relevant needs assessments but then are intending to combine

findings from two different cities without any further reflection on whether these

combined findings will remain locally relevant.

I also think it needs to be clear why you are defining healthcare priorities using

qualitative methodology now. Would it not make sense to determine who is likely to

have a poor treatment outcome first (from your quantitative work) – then purposively

sample those individuals you have identified as being at higher risk of poor treatment

outcomes.

Reply: We do not agree with the reviewer that the results are combined. Tables 1-3

present results for the two cities independently and then in a column representing the

full study cohort from the two cities. Thus, the reader can view both sets of results.

Likewise, in the text, the results for the two cities are presented and we highlight that

they were similar, lines 214-217. We did not find any difference that merit an

independent presentation or discussion. The same applies to the qualitative results,

the results were similar, had we found any difference between the two cities we would

have presented it in the results, but it was not the case.

Further reply: If this is the case perhaps a sentence to state there were no significant differences in findings between Leceister and Birmingham would be sufficient but just eyeballing there seems to be a much greater percentage of patients in Birmingham in IMD decile 1-2 than Leciester 73.6% versus 37.7%. Also Leicester cohort had no homeless or imprisoned patients included.

Reply: In line 266-267 of the results section we state that the results were similar in both cities “the multivariate analysis with poor treatment outcome demonstrated that in both settings” the reason we did not comment on the possible difference at baseline in IMD is because it did not have any effect in the primary outcome of the quantitative analysis which was our focus.

Regarding the use of qualitative methods to define healthcare priorities. This is simply

something that cannot be done using only quantitative methods. In lines 121-123 of the

introduction, we mention that “Most studies on TB treatment outcomes in underserved

populations in England have been done in London, however this may not be

representative of the national picture” For this reason, in terms of TB control, the

epidemiology of TB in London has driven control policy in England based mainly on a

few quantitative studies, and this is exactly what this study aims to address. We also

mentioned in lines 114-116 of the introduction that “intervention appropriateness

should be informed by a local needs assessment. For this, the target population must

be first defined and characterised to enable the identification of their healthcare

priorities.”

Further Reply: I think there is a bit of confusion here. I am not suggesting that you define the healthcare priorities using quantitative methods but that purely you could have used it to facilitate purpose sampling.

Reply: We agree with the reviewer that the quantitative analysis could have been used to inform purpose sampling, and when we were conceiving the study we thought to do it this way, but it was very difficult to operationalise, and our stakeholders from the National Health Service and the UK Health Security Agency needed to gather this evidence for decision making. For this reason, we opted for this approach which allowed us to gather information more readily. 

On the selection of participants for the qualitative study, we think that both approaches

are valid, the one the reviewer proposed and the one we used. We selected patients

based on variables known to be related to poor treatment outcome and diagnostic

delay. The process was guided by the local lead TB nurses who were very familiar with

their patient populations. In fact, had we followed the approach suggested by the

reviewer we would most probably have ended up interviewing the same patients

because we were constrained by patients willingness to be interviewed, this was a

major challenge. According to reviewer’s comment we added a mention of this

limitation in the discussion section lines 508-510

Further reply: Yes agreed patients willingness to be interviewed is certainly challenging and a constraint that has to be worked within. I do think that you need to recognise that there are biases to the sampling strategy led by a TB nurse given their specific role in this context. This is part of the COREQ guidance.

Reply: We added a new statement mentioning this aspect as a limitation in lines 525-527.

METHODS –

Point 9: As mentioned above I am not sure what the justification is for performing this

work concurrently and why this has been done and what this adds.

Reply: Our answer to point 8 address this point.

Further reply: This has not been justified yet. Just a short statement as to what is the advantage of performing this concurrently would do? such as - Did you gain data in realtime to inform your topic guides and subsequently shape your interviews?

Reply: A statement on this is already in the article lines 519-525 “Capturing or assessing social vulnerability is difficult to do, and we were constrained by the approach used by the UKHSA of measuring only certain SRFs. Vulnerability is not explained by one social or demographic factor alone; it is influenced by multiple and multidimensional factors. We attempted to overcome this limitation by integrating qualitative and quantitative data to comprehend patients’ and service providers’ perspectives on the hurdles for getting a diagnosis and completing treatment”

Point 16: Table 1 – this details that 40% of the total patients had extrapulmonary TB –

at no point has there been any discussion about diagnosis of pulmonary versus

extrapulmonary and clinical complexity. This is vital if you are going to discuss and

select patients based on diagnostic delay. I appreciate this doesn’t appear to be what

this article set out to look at and that is why is seems somewhat confused.

Reply: We addressed the points related to diagnostic delay previously in points 1 and

10.

We think that a discussion on the diagnosis of pulmonary versus extrapulmonary

TB is out of the scope of this study.

Further reply: If you are going to touch on diagnostic delay in the context of TB presenting to a primary care setting. You need to put in a sentence that recognises that there is diagnostic complexity here. Especially when you are including extrapulmonary TB in your data set. If you feel it is outside of the scope of this article then I would remove the extra-pulmonary data.

Reply: We agree with the reviewer, we highlighted this limitation in lines 518-519.

Point 20: P19 – line 290-292 is it realistic/ sensible to expect TB (in a low incidence

setting) to be an immediate diagnosis that a GP considers when a patient attends with

symptoms (the majority of patients attend GP surgeries with symptoms). You should at

a minimum state – symptoms consistent with a diagnosis of TB. Furthermore,

recognise that these can be non-specific at initial presentation and that there can be

clinical complexity. It is also not clear how this relates to your primary line of

questioning for this paper. The sample here is biased as you stated in your methods

that you were selecting patients based on having had a diagnostic delay and I think

you need to justify why this was done.

Reply: The study was done in high incidence areas in Birmingham and Leicester, like

in some boroughs of London, the incidence of TB is very high in some areas of these

two cities. For this reason, the goals of the collaborative Collaborative Tuberculosis

Strategy for England 2015–2020 included to improve access to services and ensure

early diagnosis and to ensure an appropriate workforce to deliver TB control, further

information can be accessed in reference 5. Within this context, the NHS and the

UKHSA started a programme to train GPs in these areas to recognise and diagnose

TB. Thus, we think it is pertinent and relevant that TB patients suggest that GPs in

these areas should be able to recognise and diagnose TB. The results of this study

have already informed the strengthening of training for GP in high incidence areas of

Birmingham and Leicester led by the NHS and the UKHSA.

Further reply: Can I please clarify are you suggesting that a patient presenting with any symptom (not symptoms consistent with TB) –low mood, vaginal bleeding, infected CS scar, traumatic injuries, toothache, haemorrhoids, eczema etc etc – the list can continue. Should immediately be considered for a diagnosis of TB? I do agree with you we need to think about this more and in high incidence settings, but I think stating symptoms consistent with TB would be a much more sensible considered approach. I think you need to change the wording of page 37 line 348-350

Reply: We agree with the reviewer, a new sentence was added in lines 348-349 as recommended.

Furthermore, are the GP surgeries you are referring to in these data practicing in the areas of high incidence you are referring to? Are you able to back this up with your data?

Reply: We did not have data on GP registration for the patients that were interviewed, however we do know that they resided in high incidence areas and normally the GP practice is close to the place of residence.

4.DISCUSSION –

Point 22: Birmingham and Leicester are two different settings however findings have

been amalgamated and no consideration has been given to their differences in terms

of interpretation of results. You have in your introduction highlighted the importance of

local relevant data so this seems at odds with what you have done.

Reply: Although Birmingham and Leicester are different settings, they share similar

characteristics in terms of TB epidemiology and demographic make-up with large

communities with heritage from the Indian subcontinent and TB concentrated in non-

UK born individuals. We included a brief mention about this in the revised version line

127 of the introduction and cited reference 1. Likewise, we highlight in the results lines

261-263 that the qualitative results in both settings were similar and for that reason

they’re not presented separately.

Further reply: As stated above there are differences in percentages of homeless and imprisoned patients between the two cities and these are underserved groups you are highlighting in your introduction as important.

Reply: As we stated in point 8, any baseline difference did not affect the primary outcome. We already mentioned in the limitations in lines 505-506 that the TB register in Leicester did not collect information on homelessness or imprisonment.

Point 26: The reported qualitative findings don’t meet Consolidated criteria for reporting

qualitative research (COREQ) reporting guidelines.

https://academic.oup.com/intqhc/article/19/6/349/1791966

Reply: The items that we forgot to mention and that the reviewer spotted were

introduced in the revised

Further reply: The reporting doesn’t currently meet all COREQ reporting guidance please see checklist.

https://cdn.elsevier.com/promis_misc/ISSM_COREQ_Checklist.pdf

Reply: We added the missing information on the COREQ criteria to the supporting information and highlighted it in the main text lines 216-218

---

## [Editor Report · Decision Letter 2]

5 Feb 2023

Vulnerability and tuberculosis treatment outcomes in urban settings in England: a mixed-methods study.

PONE-D-22-28286R2

Dear Dr. Berrocal-Almanza,

We’re pleased to inform you that your manuscript has been judged scientifically suitable for publication and will be formally accepted for publication once it meets all outstanding technical requirements.

Kind regards,

Tom E. Wingfield

Academic Editor

PLOS ONE
---

## [Editor Report · Acceptance letter]

9 Aug 2023

PONE-D-22-28286R2 

Vulnerability and tuberculosis treatment outcomes in urban settings in England: a mixed-methods study. 

Dear Dr. Lalvani:

I'm pleased to inform you that your manuscript has been deemed suitable for publication in PLOS ONE. Congratulations! Your manuscript is now with our production department. 

Kind regards, 

on behalf of

Dr. Tom E. Wingfield 

Academic Editor

PLOS ONE